# Cryo-EM analyses reveal the common mechanism and diversification in the activation of RET by different ligands

Jie Li[1†], Guijun Shang[2†], Yu-Ju Chen[3], Chad A Brautigam[1,4], Jen Liou[3], Xuewu Zhang[1,2*], Xiao-chen Bai[1,5*]

[1]Department of Biophysics, University of Texas Southwestern Medical Center, Dallas, United States; [2]Department of Pharmacology, University of Texas Southwestern Medical Center, Dallas, United States; [3]Department of Physiology, University of Texas Southwestern Medical Center, Dallas, United States; [4]Department of Microbiology, University of Texas Southwestern Medical Center, Dallas, United States; [5]Department of Cell Biology, University of Texas Southwestern Medical Center, Dallas, United States

*For correspondence:
xuewu.zhang@utsouthwestern.edu (XZ);
Xiaochen.Bai@UTSouthwestern.edu (X-B)

[†]These authors contributed equally to this work

**Abstract** RET is a receptor tyrosine kinase (RTK) that plays essential roles in development and has been implicated in several human diseases. Different from most of RTKs, RET requires not only its cognate ligands but also co-receptors for activation, the mechanisms of which remain unclear due to lack of high-resolution structures of the ligand/co-receptor/receptor complexes. Here, we report cryo-EM structures of the extracellular region ternary complexes of GDF15/GFRAL/RET, GDNF/GFRα1/RET, NRTN/GFRα2/RET and ARTN/GFRα3/RET. These structures reveal that all the four ligand/co-receptor pairs, while using different atomic interactions, induce a specific dimerization mode of RET that is poised to bring the two kinase domains into close proximity for cross-phosphorylation. The NRTN/GFRα2/RET dimeric complex further pack into a tetrameric assembly, which is shown by our cell-based assays to regulate the endocytosis of RET. Our analyses therefore reveal both the common mechanism and diversification in the activation of RET by different ligands.
DOI: https://doi.org/10.7554/eLife.47650.001

## Introduction

Signaling through RET plays essential regulatory roles in the development of the nervous system and kidney (*Ibáñez, 2013*). Dysregulation of RET signaling is linked to many human diseases, such as Hirschsprung's disease, a disorder characterized by lack of enteric ganglia in parts of the intestine, and multiple endocrine neoplasia type 2 (MEN 2A and 2B) (*Ibáñez, 2013*). Glia cell line-derived growth factor (GDNF), Neurturin (NRTN), Artemin (ARTN) and Persephin are four ligands that induce dimerization of RET through assembling 2:2:2 ternary complexes with RET and the co-receptors GDNF receptor-α family proteins (GFRα1–4) (*Ibáñez, 2013*; *Baloh et al., 1998*; *Durbec et al., 1996*; *Kotzbauer et al., 1996*; *Milbrandt et al., 1998*; *Trupp et al., 1996*). The kinase domain of dimerized RET mediates trans-phosphorylation of a number of tyrosine residues in the cytoplasmic region, leading to activation of downstream signaling pathways, including the Ras/MAP kinase cascade and the PI3 kinase/AKT pathway (*Ibáñez, 2013*). Recently, growth and differentiation factor 15 (GDF15) and GFRα-like protein (GFRAL) have been identified as a new ligand/co-receptor pair for RET (*Emmerson et al., 2017*; *Hsu et al., 2017*; *Mullican et al., 2017*; *Yang et al., 2017*). GFRAL expression is largely restricted to the area postrema and nucleus tractus solitarius in the hindbrain. Activation of RET by GDF15 and GFRAL in these regions leads to suppression of food intake and

reduction of body weight in response to environmental stresses (*Emmerson et al., 2017*; *Hsu et al., 2017*; *Mullican et al., 2017*; *Yang et al., 2017*; *Johnen et al., 2007*). More recently, GDF15-promoted metabolic adaption has been shown to have tissue protective effects during systemic inflammation caused by bacterial and viral infections (*Luan et al., 2019*). Therefore, this newly discovered GDF15/GFRAL/RET signaling pathway may be targeted for treating obesity and preventing infection-induced tissue damage. In addition, GDF15 is known to be highly expressed in placenta trophoblasts and many cancers, likely contribute to anorexia in pregnant women and cancer patients, respectively (*Fejzo et al., 2018*). The identification of the GDF15/GFRAL/RET pathway provides the mechanism and new therapeutic targets for these conditions.

Both RET and the co-receptors are expressed on the cell surface and use their extracellular domains to interact with the ligands. Several structures of the complexes between ligands and co-receptors of RET have been solved by X-ray crystallography, which all display a 2:2 architecture where each subunit in the dimeric ligands binds the second domain (D2) of one co-receptor molecule (*Hsu et al., 2017*; *Sandmark et al., 2018*; *Wang et al., 2006*; *Parkash and Goldman, 2009*). The angle between the two halves of these 2:2 complexes however differ substantially (*Hsu et al., 2017*; *Parkash and Goldman, 2009*), raising the question how such different configurations of the ligands/co-receptors are all capable of inducing the activation of the RET kinase. The extracellular region of RET contains four atypical cadherin-like domains (CLD1-4) followed by a cysteine-rich domain (CRD) (*Figure 1A*), which have all been implicated in interacting with ligands or co-receptors (*Goodman et al., 2014*; *Amoresano et al., 2005*). The crystal structure of the CLD1-2 tandem shows an unusual clamshell-like arrangement (*Kjaer et al., 2010*), whereas high-resolution structures of the other domains in the RET extracellular region are not available. The structure of the intact RET extracellular region in complex with GDNF and GFRα1 has been determined at low resolution by negative-stain electron microscopy (EM), showing an overall batwing-like shape in which the two wing-shaped RET molecules are tethered together by the dimeric GDNF/GFRα1 complex in the middle, and suggesting that the CRD is important for the dimerization and activation of RET (*Goodman et al., 2014*). This structure however does not clearly resolve the individual domains in RET or their interactions with GDNF and GFRα1. In particular, the conformation and location of the membrane proximal domain CRD in RET remain unclear. High-resolution structures of RET ternary complexes are required for addressing the questions how the different ligands and co-receptors interact with RET and whether they can lead to different downstream signaling.

Here, we report the cryo-EM structures of four extracellular 2:2:2 ligand/co-receptor/RET complexes, including GDF15/GFRAL/RET, GDNF/GFRα1/RET, NRTN/GFRα2/RET and ARTN/GFRα3/RET, at near-atomic resolution. The structures show that the extracellular region of RET adopts a 'C-clamp' shape, which is stabilized by extensive inter-domain interactions as well as binding of calcium ion at multiple sites. Due to this unique C-clamp shape, the recruitment of two RET molecules onto dimeric ligand/co-receptor complexes brings the two RET-CRDs to close proximity to promote the dimerization and trans-autophosphorylation of the intracellular kinase domain. Surprisingly, our cryo-EM results reveal that two 2:2:2 NRTN/GFRα2/RET complexes can dimerize to form a 4:4:4 complex through a novel ligand/receptor interface. We found that the 4:4:4 complex suppresses RET endocytosis. These findings suggest that, while activating RET through a common mechanism, the different ligands can use distinct interactions and mechanisms to fine-tune RET activity, which may lead to different signaling outcomes.

## Results

### Structure determination of RET extracellular ternary complexes

To understand the activation mechanisms of RET by different ligands, we successfully reconstituted four RET extracellular ternary complexes, GDF15/GFRAL/RET, GDNF/GFRα1/RET, NRTN/GFRα2/RET and ARTN/GFRα3/RET, for cryo-EM structure determination (*Figure 1—figure supplement 1*). The initial 3D reconstruction of the 2:2:2 GDF15/GFRAL/RET complex with 2-fold symmetry applied was determined at relatively low-resolution (4.0 Å), partly due to the relative movement between the two wings in the batwing-shaped complexes. To improve the resolution, we used the symmetry expansion approach as well as focused refinement with signal subtraction as described in our previous work (*Figure 1—figure supplements 2–4*) (*Bai et al., 2015*; *Zhou et al., 2015*). The resulting

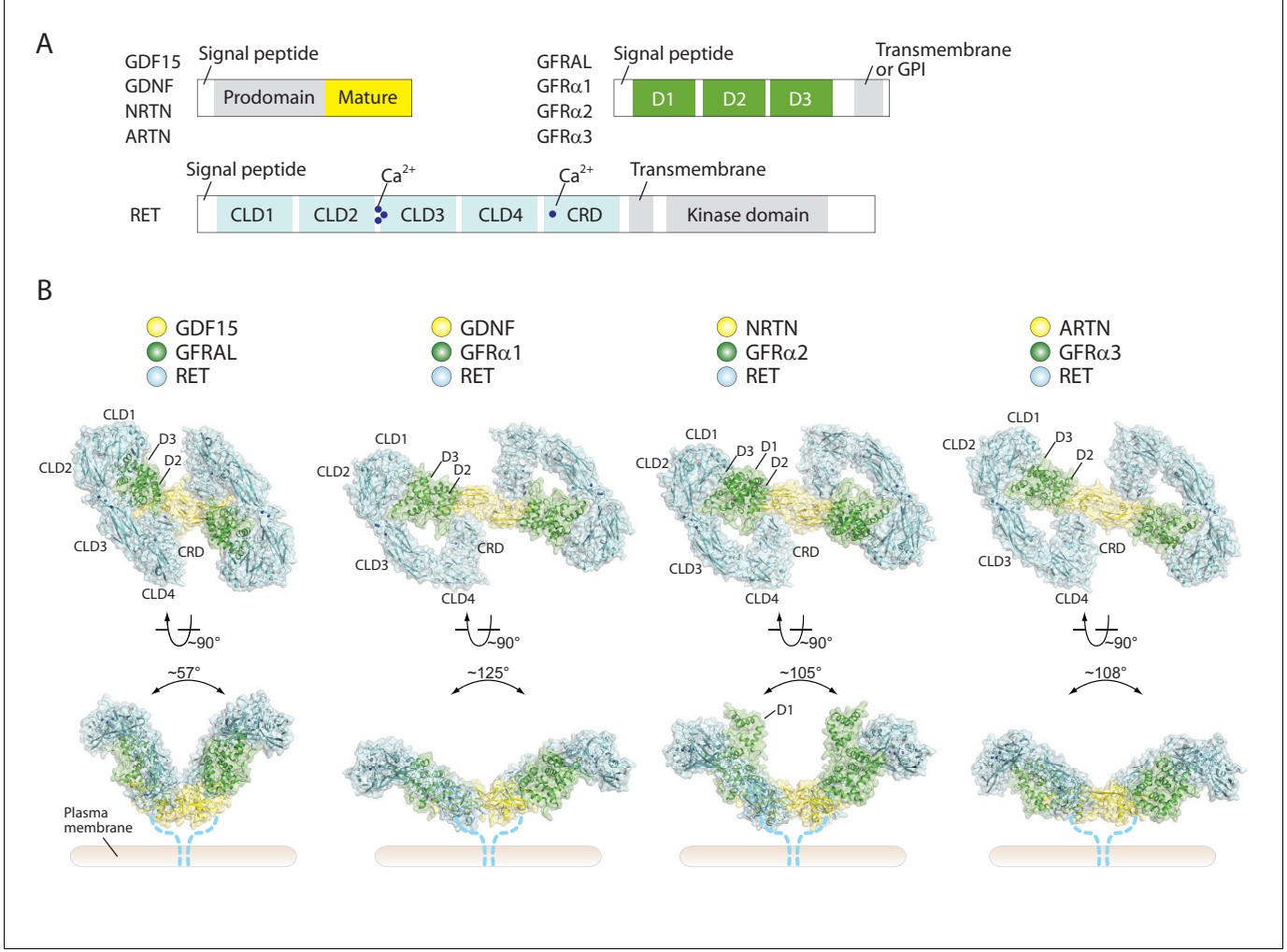

**Figure 1.** Overall structures of four different RET ternary complexes. (A) Domain organization of the ligands, co-receptors and RET. The domains in gray are absent in the cryo-EM structures. (B) Cartoon representations of the four RET ternary complexes. The angles between the two wings are indicated -in the side views. Dotted lines indicate the connection from the CRD to the transmembrane region of RET.

DOI: https://doi.org/10.7554/eLife.47650.002

The following figure supplements are available for figure 1:

**Figure supplement 1.** Purification of the four RET ternary complexes.

DOI: https://doi.org/10.7554/eLife.47650.003

**Figure supplement 2.** Flowchart of data processing.

DOI: https://doi.org/10.7554/eLife.47650.004

**Figure supplement 3.** Cryo-EM analyses of four RET ternary complexes.

DOI: https://doi.org/10.7554/eLife.47650.005

**Figure supplement 4.** Representative- cryo-EM density of various parts of the four ternary RET complexes.

DOI: https://doi.org/10.7554/eLife.47650.006

**Figure supplement 5.** Additional cryo-EM maps.

DOI: https://doi.org/10.7554/eLife.47650.007

**Figure supplement 6.** Expanded view of the different ligands/co-receptors.

DOI: https://doi.org/10.7554/eLife.47650.008

reconstruction comprising one RET and GFRAL with the GDF15 dimer bound reached 3.7 Å resolution. This reconstruction in combination with the intact map with the 2-fold symmetry allowed us to build an accurate atomic model for the entire 2:2:2 complex (*Figure 1* and *Figure 1—figure supplement 4*). The same procedure was used for the structure determination of the 2:2:2 NRTN/GFRα2/RET and ARTN/GFRα3/RET complexes, leading to reconstructions to 3.4 Å and 3.5 Å resolution,

respectively (*Figure 1—figure supplement 3*). The reconstruction of the GDNF/GFRα1/RET complex reached lower resolution (4.4 Å) due to preferred orientation of particles (see Materials and methods for details) (*Figure 1—figure supplement 3*).

## Overall architecture of RET extracellular ternary complexes

All four complexes show the same batwing-like architecture as shown in the previously reported negative-stain EM structure of the GDNF/GRFα1/RET (*Goodman et al., 2014*), with the dimeric ligands located at the center connecting the two wings formed together by RET and the co-receptors (*Figure 1*). A recently published medium resolution (5.7 Å) cryo-EM structure of the NRNT/GFRα2/RET complex displays a similar overall architecture (*Bigalke et al., 2019*). The outer edge of the wing is defined by RET, which assumes an overall 'C'-shape to clamp on both the ligands and co-receptors. Low-resolution reconstructions of RET in the apo-state with cryo-EM by us (*Figure 1—figure supplement 5C*) and with small-angle X-scattering reported previously both show a similar C-clamp shape (*Goodman et al., 2014*), suggesting that the RET extracellular region is relatively rigid and does not undergo substantial conformational changes upon binding of the ligands/co-receptors. Extensive interactions are formed between the consecutive domains in the RET extracellular region, explaining this rigidity (*Figure 2A–2D*). As expected, all the four CLDs in RET adopt the cadherin-like fold characterized by a two-layered β-sandwich. CLD1 and CLD2 pack tightly against each other to form the clamshell-like structure as shown in the published crystal structure of these two domains (*Figure 2A*) (*Kjaer et al., 2010*). CLD3 is connected to CLD2 with an inter-domain angle of ~ 144°, which is maintained by binding of three calcium ions at the inter-domain junction (*Figure 2B*). The calcium ions are coordinated by a number of conserved negatively charged residues from CDL2 and CLD3 that have been identified as classical cadherin calcium-binding motifs and important for the correct folding of RET (*Figure 2B*) (*Goodman et al., 2014*; *van Weering et al., 1998*; *Kjaer and Ibáñez, 2003*; *Anders et al., 2001*). The inter-domain angle between CLD3 and CLD4 is ~ 130°, leading to a smooth curved shape of CLD2, CLD3 and CLD4. The interface between CLD3 and CLD4 is dominated by hydrophobic residues (*Figure 2C*). CRD docks onto CLD4 in an orthogonal orientation through an extensive interface containing both hydrophobic and polar interactions, completing the bottom portion of the C-clamp (*Figure 2D*).

## Structure of CRD in RET

Notably, CRD in RET adopts a new fold that does not show significant similarity to other proteins in the PDB database. This small globular domain is composed of three small 2-or 3-stranded β-sheets and two short helices, which are connected by extensive loops (*Figure 2E and F*). These structural elements are stapled together by seven disulfide bonds. Cys630 and Cys634 in the linker between CRD and the transmembrane region of RET have been shown to form another disulfide bond, but the corresponding density is not well-resolved in our cryo-EM maps (*Chappuis-Flament et al., 1998*). There is a strong blob of density in a pocket surrounded by a number of conserved acidic residues in CRD, including Asp567, Glu574 and Asp584, as well as several backbone carbonyl groups (*Figure 2E and H*). The arrangement of the oxygen atoms around the site resembles closely the calcium binding site in calmodulin (*Figure 2G*). We therefore tentatively assign this as a new calcium binding site in RET.

The structure of the RET extracellular region provides a basis for understanding disease-causing mutations of some cysteine residues in RET-CRD. For example, mutations of Cys609, Cys611, Cys618, Cys620 and Cys634 cause MEN2A (*Figure 2—figure supplement 1*) (*Mulligan et al., 1993*; *Mulligan, 2014*). These cysteine residues sit either at the periphery of CRD or in the linker between CRD and the transmembrane region of RET (*Figure 2E* and *Figure 2—figure supplement 1*). Mutations of these residues may not dramatically affect the folding or cell surface localization of RET, but leave one unpaired cysteine residue at the periphery of the protein, which can subsequently induce ligand-independent dimerization and constitutive activation of RET by forming an inter-chain disulfide bond (*Chappuis-Flament et al., 1998*; *Kjaer et al., 2006*). On the other hand, loss-of-function mutations throughout the RET extracellular region have been associated with Hirschsprung's disease (*Edery et al., 1994*; *Pelet et al., 1998*; *Iwashita et al., 1996*). Many of these mutations are distributed in the core of the protein, the co-receptor binding site or the calcium binding sites, thus likely

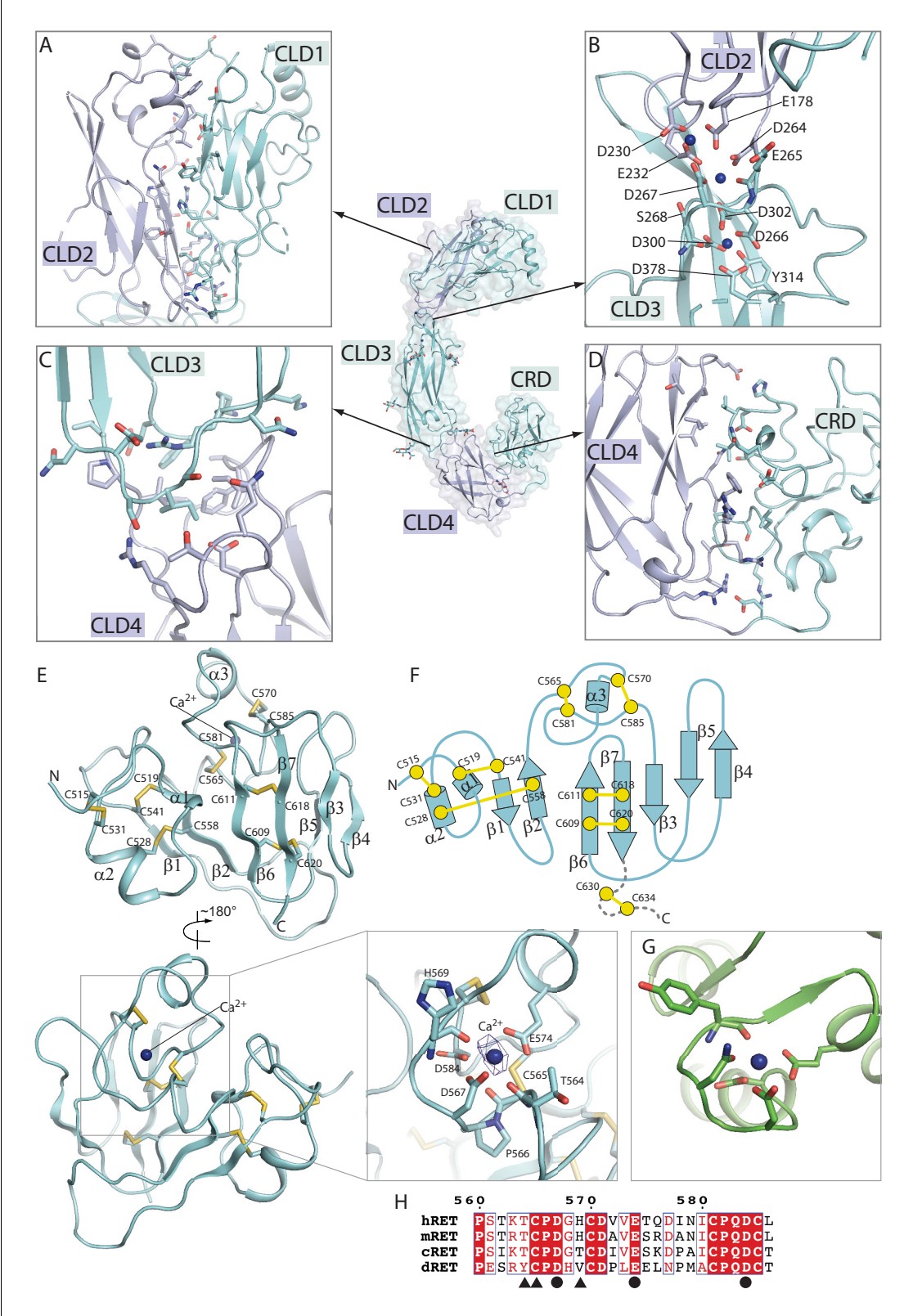

**Figure 2.** Structure of the RET extracellular domain. (**A–D**) Overall structure and the inter-domain interactions in the RET extracellular region. (**E**) Structure of the CRD domain of RET. The expanded view shows the details of the putative calcium binding site. The density for the calcium ion is very strong, displayed as blue mesh at the 20σ threshold. (**F**) Topology diagram of the CRD domain. (**G**) Calcium binding site in calmodulin (PDB ID: 1cll). *Figure 2 continued on next page*

*Figure 2 continued*

(H) Sequence alignment of the calcium binding segment in the CRD domain of RET from human (h), mouse (m), chicken (c) and *Drosophila* (d). Circles and triangles highlight residues coordinating the calcium ion with sidechains and backbone carbonyl, respectively.
DOI: https://doi.org/10.7554/eLife.47650.009
The following figure supplement is available for figure 2:

**Figure supplement 1.** Mapping of the disease-associated point mutations onto the RET extracellular domain.
DOI: https://doi.org/10.7554/eLife.47650.010

affecting the stability or co-receptor binding of RET (*Manié et al., 2001*) (*Figure 2—figure supplement 1*).

## Inter-subunit interactions in the RET ternary complexes

In all of the four ternary complex structures, the ligands use the convex face formed mostly by the two 'finger' loops to bind to the second domain (D2) of their respective co-receptors, similar to those shown by the previously determined X-ray structures of the ligand/co-receptor complexes (*Hsu et al., 2017*; *Sandmark et al., 2018*; *Wang et al., 2006*; *Parkash and Goldman, 2009*); therefore, we will not describe the ligand/co-receptors interface in detail here.

RET makes contacts with the co-receptors and ligands by its N-terminal and C-terminal domains, respectively. The D3 domain of the co-receptors forms bipartite interfaces (denoted as interface I and II) with the N-terminal portion of the RET C-clamp in all of our reconstructions, although the detailed interactions are different in each (*Figure 3A*, *Figure 3—figure supplement 1A, C and E*). In the GDF15/GFRAL/RET complex, interface I is formed between a helix-loop-helix-loop motif (residues 246–265) in the D3 domain of GFRAL and the concave surface on CLD1 and CLD2 in RET (*Figure 3A*). The interface includes Trp37, Ala35, Thr120, Tyr146 and Thr170 from RET, Lys251, Thr261 and Ser263 from GFRAL. Interface II is made between a short loop (residues 295–299) in GFRAL and the CLD2-CLD3 junction near the calcium binding sites (*Figure 3A*). The segments in GFRα1, GFRα2 and GFRα3 corresponding to the helix-loop-helix-loop motif in GFRAL are much shorter and adopt extended loop conformations (*Figure 3—figure supplement 1A, C and E*). As a result, interfaces I in GFRα1/RET, GFRα2/RET and GFRα3/RET are much smaller (buried surface areas less than 700 Å$^2$) and likely weaker than that in GFRAL/RET (buried surface area ~ 1500 Å$^2$). Interfaces II of GFRα1, GFRα2 and GFRα3 with RET (buried surface areas ~ 730–1010 Å$^2$) however are more extensive than that in GFRAL/RET (buried surface areas ~ 680 Å$^2$), which may partially compensate for their weaker interfaces I. The D1 domain of the co-receptors is invisible in the 3D reconstructions of GDF15/GFRAL/RET, GDNF/GFRα1/RET, and ARTN/GFRα3/RET, suggesting that it is not involved in the binding to either RET or the ligands in these three complexes. In contrast, the D1 domain of GFRα2 is resolved in the cryo-EM map of the NRTN/GFRα2/RET complex, and packs closely with the D3 domain as seen in the crystal structure of the NRTN/GFRα2 complex (*Sandmark et al., 2018*). The D1 domain of GFRα2 makes a few contacts with RET-CLD1, which may help stabilize the ternary complex.

At the C-terminal portion of the RET C-clamp, RET-CRD interacts directly with the concave surface of the finger loops in the ligands, opposite to the side that binds the co-receptors (*Figure 3B*). As a result, the finger loops in the ligands appear to wedge between RET-CRD and the co-receptors. In the GDF15/GFRAL/RET complex, GDF15 embraces the surface formed by strands β3–7 along with several inter-strand loops in RET-CRD, through mainly hydrophobic interactions, burying Trp228, Met253 and Tyr297 in GDF15 and Ile551, Val591, Gly593, Tyr606 and Phe619 of CRD (*Figure 3B*). The interface also contains polar residues, such as Gln247 and Gln256 in GDF15. Similar ligand/CRD interfaces are formed in the other three ternary complexes (*Figure 3—figure supplement 1B, D and F*).

The simultaneous binding of RET to both the ligand and co-receptor explains the cooperativity in the formation of the ternary complexes, while neither the ligand nor co-receptor alone is sufficient for activation of RET (*Schlee et al., 2006*). To confirm the binding mode as shown in our structures, we introduced a T261R mutations in GFRAL to disrupt its interface I with RET. Our pull-down assay results showed that wild-type GFRAL was able to pull down RET and GDF15 simultaneously, indicating the formation of the ternary complex (*Figure 3C*). In contrast, the T261R mutant of GFRAL failed

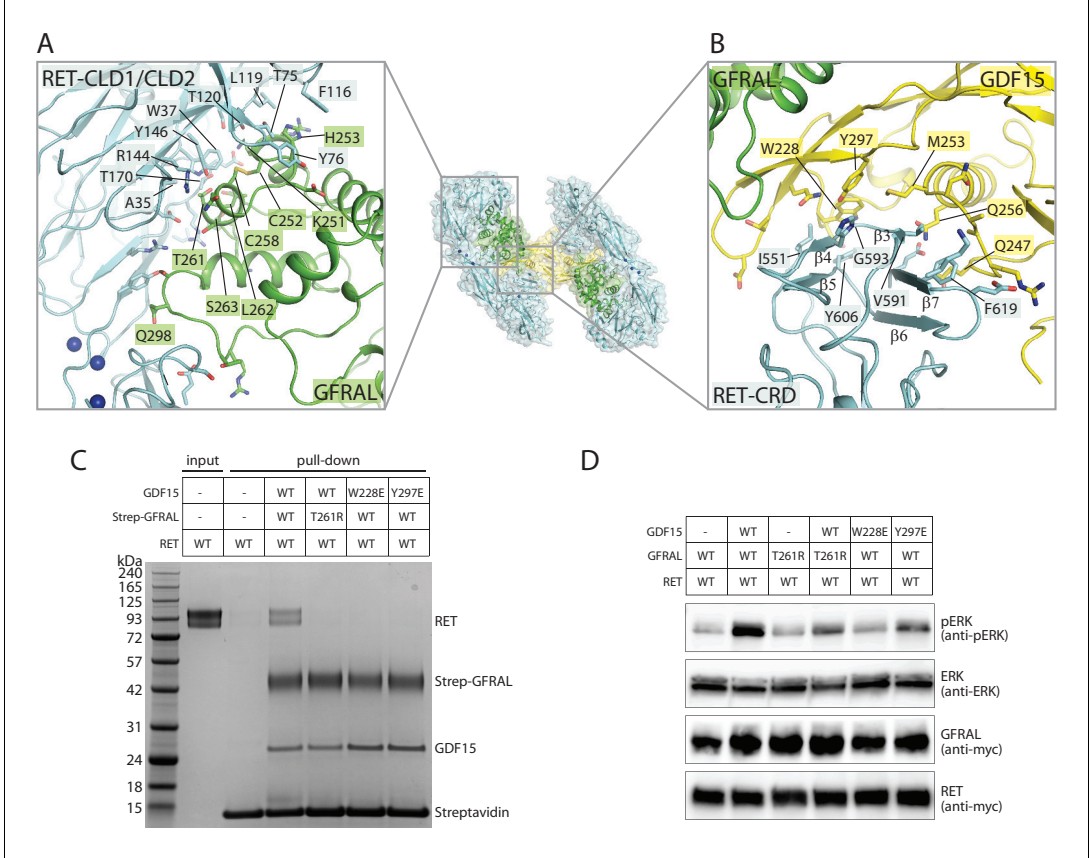

**Figure 3.** Binding interfaces in the GDF15/GFRAL/RET complex. (A) Interface between RET and GFRAL. An overall view of the complex structure is shown in the middle as a reference. (B) Interface between RET-CRD and GDF15. (C) Pull-down assays for the GDF15/GFRAL/RET ternary complex. Strep-tagged GFRAL bound to Streptavidin-conjugated beads was used to pull down GDF15 and RET. Streptavidin beads without Strep-GFRAL bound served as negative control. Mutations in either GFRAL or GDF15 abolish binding of RET. The results shown are representative of three biological repeats. (D) GDF15-induced ERK phosphorylation in HEK293 cells expressing full-length myc-tagged RET and GFRAL. Cells were treated with GDF15 at 10 nM for 15 min. ERK phosphorylation levels (pERK) were assessed by western blot. Expression levels of RET and GFRAL were monitored by anti-myc western blot. The results shown are representative of three biological repeats.

DOI: https://doi.org/10.7554/eLife.47650.011

The following figure supplement is available for figure 3:

**Figure supplement 1.** Binding interfaces in the GDNF/GFRα1/RET, NRTN/GFRα2/RET and ARTN/GFRα3/RET complexes.

DOI: https://doi.org/10.7554/eLife.47650.012

to support the formation of the GDF15/GFRAL/RET ternary complex (*Figure 3C*). Similarly, the mutations W228E and Y297E in GDF15, which target the interaction with RET-CRD, also abolished the formation of the ternary complex (*Figure 3C*). Furthermore, we examined the effects of these mutations on RET signaling by monitoring ERK phosphorylation in cells. Treatment of cells stably expressing both RET and GFRAL with wild-type GDF15 induced robust phosphorylation of ERK (*Figure 3D*). However, the W228E and Y297E mutations in GDF15 led to dramatically decreased phosphorylation of ERK (*Figure 3D*). Similarly, cells expressing wild-type RET but the T261R mutant of GFRAL showed lower levels of ERK phosphorylation when stimulated with wild-type GDF15 (*Figure 3D*). These results together confirm that the interfaces as seen in the structure are important for the formation of the GDF15/GFRAL/RET ternary complex as well as the activation of RET in cells.

## Activation mechanism of RET

In all of the four ternary complex structures, the ligand dimer directly interacts with the two CRDs from the two RET molecules and thereby bring them into close proximity. In addition, clear but

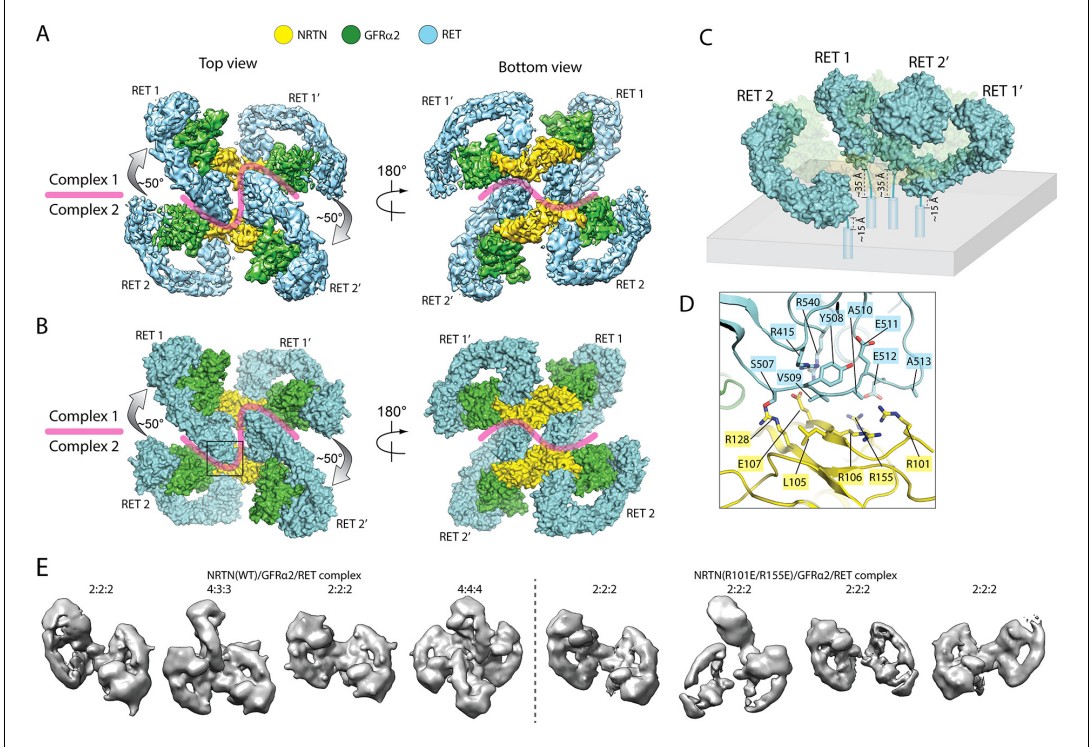

**Figure 4.** Higher-order oligomerization of the NRTN/GFRα2/RET complex. (**A**) Refined map of the 4:4:4 NRTN/GFRα2/RET complex. (**B**) Atomic model of the 4:4:4 NRTN/GFRα2/RET complex shown in the surface representation. (**C**) Distance of the four RET molecules in the 4:4:4 complex to the plasma membrane. The gray box represents the plasma membrane. NRTN and GFRα2 are rendered semi-transparent to clearly show the position of the four RET molecules relative to the membrane. (**D**) Detailed view of the new interface between NRTN and RET that mediate the formation of the 4:4:4 complex. The view is expanded from the boxed region in the left panel of (**B**). (**E**) 3D class averages of the NRTN(WT)/GFRα2/RET and NRTN(R101E/ R155E)/GFRα2/RET complexes. Wild-type NRTN formed complexes with RET and GFRα2 larger than the 2:2:2 stoichiometry, which were eliminated by the R101E/R155E mutation.

DOI: https://doi.org/10.7554/eLife.47650.013

The following figure supplement is available for figure 4:

**Figure supplement 1.** Model building procedure for the 4:4:4 NRTN/GFRα2/RET complex.
DOI: https://doi.org/10.7554/eLife.47650.014

relatively weak cryo-EM densities at the bottom faces of the ligands were observed for the linker between CRD and the transmembrane region of RET (*Figure 1—figure supplement 5A and B*). These observations suggest that the two copies of the linker converge near the plasma membrane. This configuration of the two RET molecules is poised to induce the dimerization of the transmembrane region and intracellular kinase domain, leading to cross-autophosphorylation of RET and subsequent activation of the downstream signaling cascades. Therefore, all the ligand/co-receptor pairs use the same general mechanism to activate RET. However, there are large differences in the angle between the two wings of the batwing-shaped complexes (*Figure 1B*), which mainly originate from the different conformations of the ligand dimers and their different interfaces with the co-receptors as noted previously (*Figure 1—figure supplement 6*) (*Hsu et al., 2017*; *Sandmark et al., 2018*; *Parkash and Goldman, 2009*). This angle in the GDF15/GFRAL/RET complex (~60°) is much more acute than that in the complexes of GDNF/GFRα1/RET (130°), NRTN/GFRα2/RET (105°) and ARTN/ GFRα3/RET (108°) (*Figure 1B*). Associated with this angular variation, the distances between two CRDs in the complexes also differ to some extent. It is appealing to speculate that the differences in the angle and distance between the two RET molecules in the complexes may provide a mechanism in defining RET signaling specificity, which could potentially allow the five different ligands of RET to generate distinct signaling outputs from the same but versatile receptor.

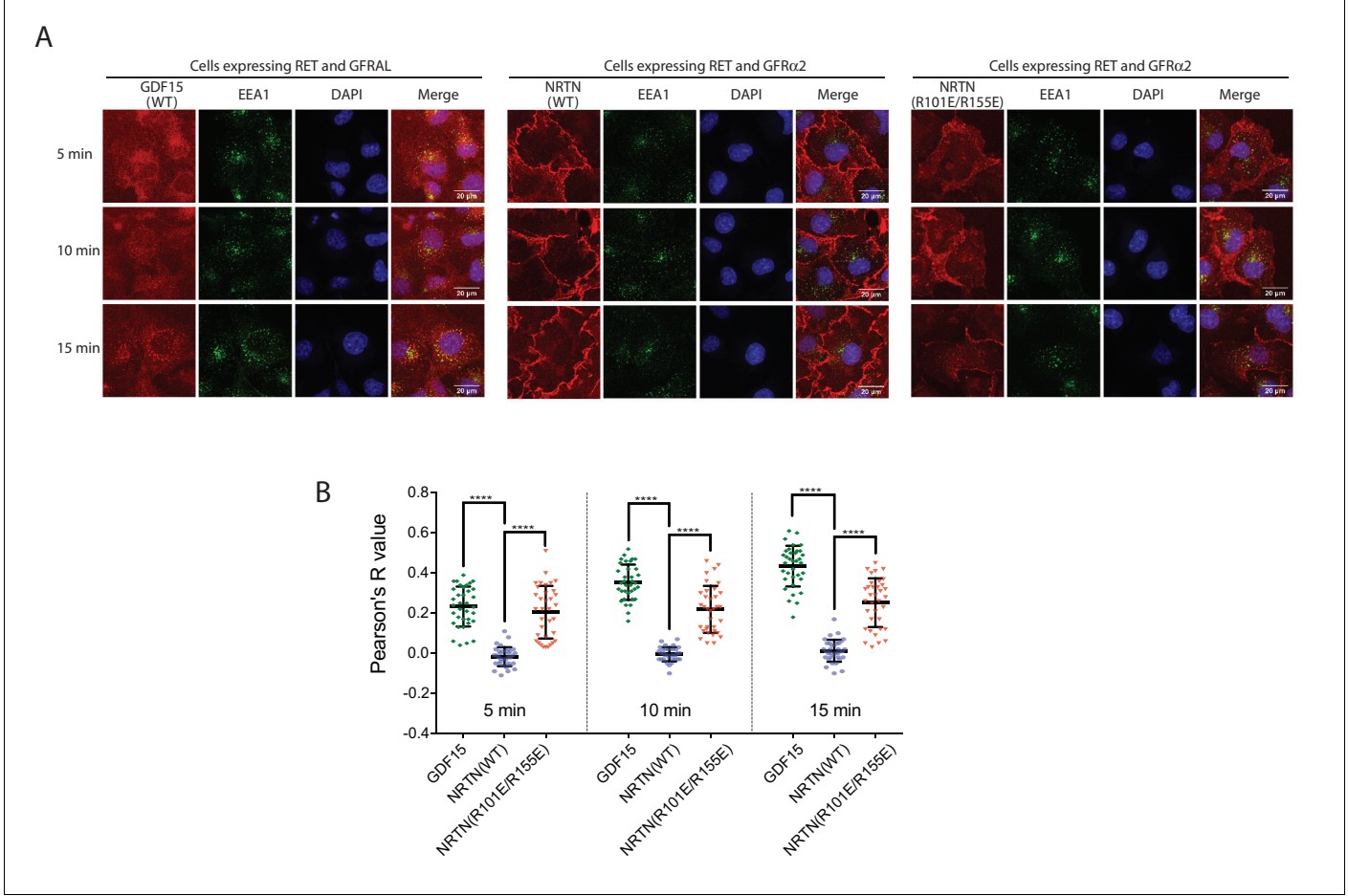

**Figure 5.** The 4:4:4 NRTN/GFRα2/RET complex delays RET endocytosis. (**A**) Endocytosis of the NRTN/GFRα2/RET and GDF15/GFRAL/RET complexes. Fluorescently labeled NRTN (wild-type or the R101E/R155E mutant) and GDF15 were incubated with COS7 cells expressing GFRα2/RET and GFRAL/RET, respectively, and imaged at indicated time points. EEA1 were immuno-stained to serve as an early endosome marker. (**B**) Quantification of the colocalization of NRTN wild-type, the R101E/R155E mutant and GDF15 with EEA1. Pearson's correction coefficients between the ligands and EEA1 were calculated for 35 cells in each group from two biological repeats. Each dot in the scatter plot represent one cell. The bars represent mean and standard deviation. P-values were calculated using the two-tailed Welch's t-test. ****p<0.0001.

DOI: https://doi.org/10.7554/eLife.47650.016

The following source data is available for figure 5:

**Source data 1.** Source data for *Figure 5B*.
DOI: https://doi.org/10.7554/eLife.47650.017

## RET can form higher-order oligomeric complexes

Surprisingly, some 2D class averages of the NRTN/GFRα2/RET and ARTN/GFRα3/RET complexes appear to be larger than the 2:2:2 complexes. With further 3D classification for the dataset of the NRTN/GFRα2/RET complex, we identified one class showing two 2:2:2 complexes arranged in a staggered fashion to form a 4:4:4 complex that resembles a four-bladed propeller. Subsequent 3D refinement yielded a map at 4.3 Å resolution, allowing us to build a complete model of the 4:4:4 complex by rigid-body fitting the atomic models of the protein components into cryo-EM map (*Figure 4*, *Figure 4—figure supplement 1* and *Video 1*). We performed the same image processing procedure for the dataset of the ARTN/GFRα3/RET complex, but the 3D reconstruction failed due to inadequate number of particles.

Simple modeling of the 4:4:4 NRTN/GFRα2/RET complex on the surface of the plasma membrane shows that the distances of the four RET molecules to the membrane surface are different, due to a relative rotation of ∼ 50° between the two 2:2:2 complexes (*Figure 4C*). As shown in *Figure 4C*, the

distance between the end of the CRD in the two centrally located RET molecules (RET1 and RET2′) and the membrane is ~ 35 Å, whereas that of the two peripheral RET molecules (RET2 and RET1′) is ~ 15 Å. The linker (residues 623–636) between the last residue in RET-CRD and the N-terminus of transmembrane region is 14-residue long, but its linear span is shortened by the disulfide linkage between Cys630 and Cys634 to approximately equivalent to a 11-residue linker. This linker can readily bridge the 15 Å distance for the peripheral RET2 and RET1′, but may be stretched to some extent in order to span the 35 Å distance for the central RET1 and RET2′. These analyses together suggest that the 4:4:4 complex can be formed by full-length RET on the cell surface, but it may display preference for membrane areas with a positive curvature.

The 4:4:4 NRTN/GFRα2/RET complex is mediated by a novel interface, formed between the protruding 'elbow' at the CLD4-CRD junction of RET and the convex face of NRTN near the GFRα2-binding site (*Figure 4D*). Particularly, the loop connecting CLD4 and CRD (residues 507–513) in RET makes a number of interactions with the surface of the major β-sheet in NRTN. Several charged residues in NTRN, including Arg101, Arg106, Glu107, Arg128, and Arg155, contribute to the interaction. We introduced an R101E/R155E double mutation to NRTN, and imaged the NRTN(R101E/R155E)/GFRα2/RET complex with cryo-EM. 3D classification result of this mutant complex clearly showed that the double mutations in NRTN exclusively abolished the formation of the 4:4:4 complex, but not the 2:2:2 complex, verifying that the novel NRTN/RET interface is responsible merely for the formation of the higher-order oligomer (*Figure 4E*).

In the 4:4:4 NRTN/GFRα2/RET complex, two of the RET molecules (RET1 and RET2′ in *Figure 4*) are sandwiched between the two wings of the adjacent 2:2:2 complex. This configuration is only compatible with the structures of the NRTN/GFRα2/RET, GDNF/GFRα1/RET and ARTN/GFRα3/RET complexes where the angle between the two wings is large (*Figure 1B*). The angle between the two wings in the GDF15/GFRAL/RET complex is however much tighter, and therefore cannot accommodate the third RET molecule for the formation of the 4:4:4 complex. Indeed, no class averages of the 4:4:4 complex can be identified in the cryo-EM dataset of the GDF15/GFRAL/RET complex. Thus, assembling distinct signaling complexes of different oligomeric states may be a mechanism for the five ligands of RET to trigger distinct signaling outcomes through the same receptor. This difference seems to be in line with the fact that the signaling of the GDF15/GFRAL/RET complex is mostly involved in regulating metabolic responses, whereas the other RET complexes control cell growth, proliferation and survival.

## Higher-order oligomerization regulates RET - endocytosis

Ligand-induced RET internalization through clathrin-mediated endocytosis plays a role in regulating signaling (*Crupi et al., 2015*; *Richardson et al., 2012*; *Richardson et al., 2006*). We speculated that the potential preference of the 4:4:4 NRTN/GFRα2/RET complex for positive membrane curvature mentioned above might impede its entrance to clathrin-coated pits, which have negative membrane curvature. It has been shown that the geometry and size of proteins can regulate vesicle secretion or endocytosis by changing the curvature of membrane, through mechanisms that are distinct from the membrane-bending effects of the BAR domains (*DeGroot et al., 2018*; *Shurer et al., 2019*). To examine the effect of the 4:4:4 complex on RET endocytosis, we treated COS7 cells stably expressing full length RET and GFRα2 with fluorescence-labeled NRTN and monitored its endocytosis. The results showed that most wild-type NRTN remained on the cell surface as long as 15 min (*Figure 5A*). Very few intracellular puncta of NRTN were formed and co-localized with the early endosome marker EEA1 (early endosome antigen 1)

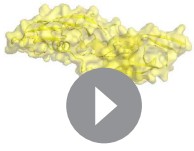

**Video 1.** The assembly of the NRTN/GFRα2/RET 2:2:2 and 4:4:4 complexes.
DOI: https://doi.org/10.7554/eLife.47650.015

(*Figure 5A*). In contrast, the NRTN(R101E/R155E) mutant were internalized quickly and formed many puncta co-localizing with EEA1 at the 5 min time point, which progressed to higher levels at 10 and 15 min (*Figure 5A*). For comparison, we carried out similar experiments for the GDF15/GFRAL/RET complex, which does not form higher-order oligomers. GDF15 underwent quick internalization, even faster than the NRTN(R101E/R155E) mutant (*Figure 5A*). We used the Pearson's correlation coefficient between the ligands and EEA1 as a quantitative indicator of the internalization of NRTN and GDF15 (*Manders et al., 1992*). The quantification results show that the colocalization of wild-type NRTN with EEA1 remained at low levels at all of the three time points (*Figure 5B*). The NRTN(R101E/R155E) mutant displayed significantly higher levels of co-localization with EEA1, but lower than that of GDF15 (*Figure 5B*). These results together support the idea that the different oligomeric states of RET induced by the ligands can regulate the internalization. The slower endocytosis of the 4:4:4 NRTN/GFRα2/RET complex allows it to maintain the active state on the cell surface for a longer period of time, which may lead to distinct signaling outcomes as compared with the GDF15/GFRAL/RET complex that undergoes fast endocytosis.

## Oligomerization of RET in the apo-state

Some RTKs such as the epidermal growth factor receptor (EGFR), insulin-like growth factor receptor and insulin receptor (*Kavran et al., 2014*; *Kovacs et al., 2015*) can form inactive dimers on the cell surface in the absence of ligand. Using analytical ultracentrifugation (AUC), we found that the extracellular region of apo-RET exhibits an intrinsic ability to oligomerize, existing in an equilibrium favoring the monomeric (4.8 S) form at the concentrations used, but evincing higher-order oligomers ranging from dimers (ca. 8 S) to perhaps tetramers or pentamers (13.5 S) (*Figure 6E*). This solution behavior comports with the appearance in apo-RET cryo-EM micrographs of elongated entities that were clearly larger than monomeric apo-RET (2D class averages are shown in *Figure 6D*). We were unable to determine the structures of these oligomers due to conformational heterogeneity. Interestingly, a crystallographic dimer of the CLD1-2 domains of RET has been reported previously (*Figure 6A*) (*Kjaer et al., 2010*). We constructed a dimer of the full-length extracellular region of RET by superimposing two protomers of our RET model with the CLD1-2 crystallographic dimer, which resembles some of the 2D class averages (*Figure 6B and D*). This dimer model is further supported by the AUC results showing that a double mutation of Arg77 and Arg144 in the dimer interface abolished the oligomerization of the RET extracellular region (*Figure 6E*). In this ligand-free RET dimer model, the C-termini of the two CRD domains point to the same direction, consistent with their connection to the transmembrane region of RET expressed on the cell surface. However, the two CRDs in this dimer are placed far apart (~120 Å), suggesting that it is in an inactive state of RET, because it imposes a large spatial separation of the two kinase domains that is unfavorable for cross-phosphorylation. The interface of the apo-RET dimer partially overlaps with the binding interface for the co-receptors (*Figure 6C*). Therefore, binding of a co-receptor to RET can both disrupt the inactive dimer and facilitate the ligand/RET interaction, eventually leading to the active dimer conformation of RET as seen in the structures of the ternary complexes.

## Discussion

Based on our structural analyses and previous studies, we propose a multi-state model for the regulation of RET signaling (*Figure 7*). This overall model and the structural details revealed by our cryo-EM structures provide a basis for understanding many mutations in RET that are associated with human diseases. The batwing-shaped dimeric complexes is the common mechanism for RET activation by the ligands and their respective co-receptors, while using quite different atomic interaction for the formation of the complex. The precise role of the apo-dimer of RET is not clear at present. It may help keep the kinase from cross-phosphorylation in the absence of the ligand. On the other hand, the existing dimer of RET on the cell membrane may facilitate the ligand-induced activation, as it enables a simple switch from the inactive to the active dimer as seen in some other RTKs such as EGFR and the insulin receptor (*Kavran et al., 2014*; *Kovacs et al., 2015*). The formation of the higher-order oligomer of the NRTN/GFRα2/RET complex suggests an unanticipated novel layer of regulation for RET. Our data suggest that the formation of the 4:4:4 NRTN/GFRα2/RET complex delays RET endocytosis, likely leading to more sustained signaling. Along this line, previous studies have shown that RET endocytosis regulates both the duration and choice of pathway of signaling

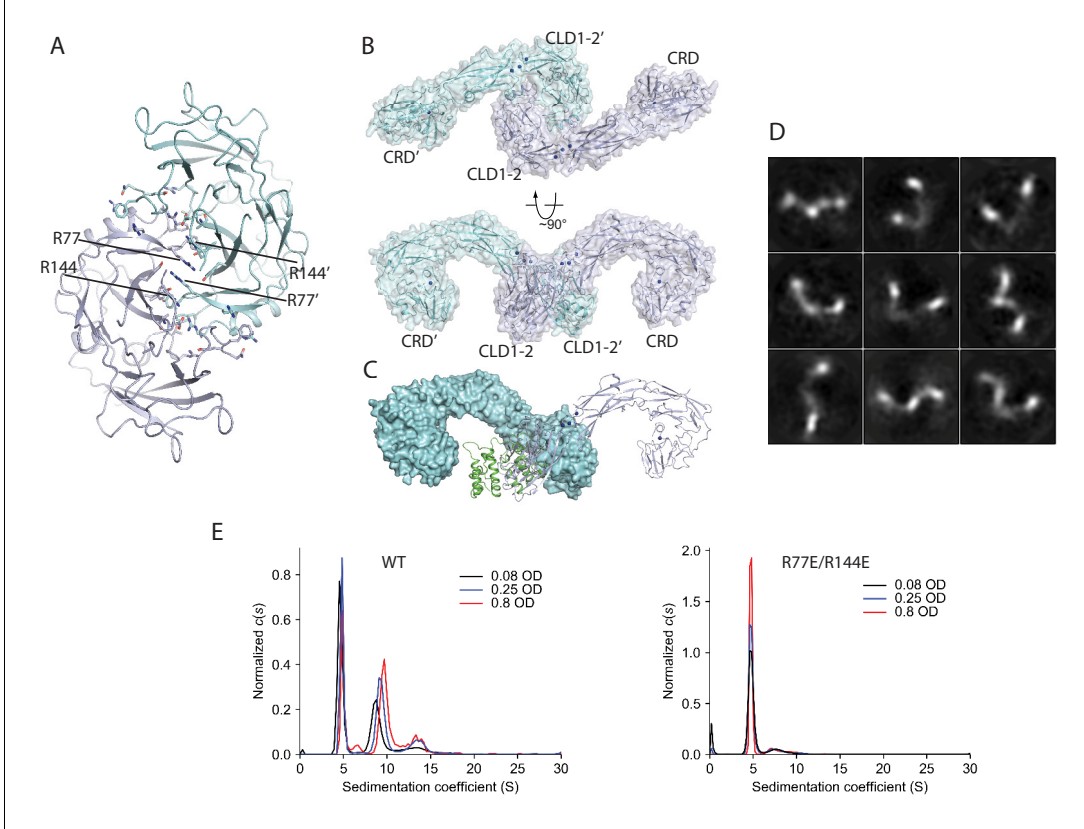

**Figure 6.** Ligand-independent dimerization of RET. (**A**) The dimer of RET-CLD1/CLD2 in a previously reported crystal structure (PDB ID: 2X2U) (**B**) The dimeric model of the full-length extracellular region of RET in the apo-state based on the dimer in (**A**). (**C**) The co-receptor binding interface overlaps with the apo-RET dimer interface. The two protomers in the RET dimer are shown in surface (cyan) and cartoon (blue) representations, respectively. GFRAL (green) bound to RET in cyan clashes with RET in blue, suggesting that binding of GFRAL can disrupt the dimer of apo-RET. (**D**) 2D class averages of RET in the apo-state. (**E**), Analysis of the oligomerization state of the apo-RET extracellular region by AUC. The peak with the sedimentation coefficient of 4.8 S corresponds to the RET monomer. Wild-type RET also contains another species with the sedimentation coefficient of ~ 7–9 S, suggesting higher-order oligomerization. The R77E/R144E mutant ran predominately as a monomer.

DOI: https://doi.org/10.7554/eLife.47650.018

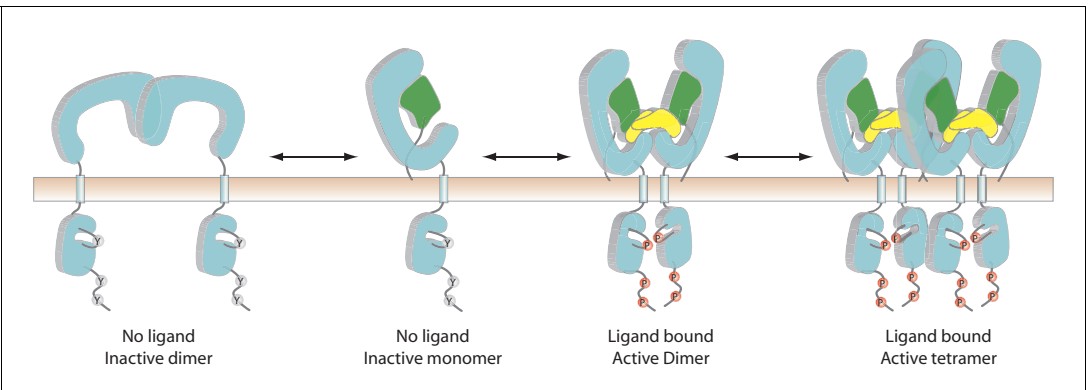

**Figure 7.** A multi-state model of RET regulation on the cell surface. RET, the co-receptor and ligand are colored cyan, green and yellow, respectively. 'Y' and 'P' represent unphosphorylated and phosphorylated tyrosine residues, respectively.

DOI: https://doi.org/10.7554/eLife.47650.019

(*Richardson et al., 2012*; *Richardson et al., 2006*). NRTN mutations have been associated with Hirschsprung's disease, implicating that the NRTN-induced signaling of RET plays a role in the growth or guidance of enteric ganglia (*Doray et al., 1998*; *Enomoto et al., 2001*). The high-order NRTN/GFRα2/RET complex may be a mechanism to maintain sustained signal of RET required for these functions.

Our analyses suggest that the formation of the high-order oligomeric complex is a specialized function of a subset of ligands of RET, as GDF15 and GFRAL cannot form similar 4:4:4 complex with RET because of the tight angle between the two wings in the dimeric complex. Therefore, while the batwing-shaped dimeric complex serves as the common basis for RET activation by all the ligands, the variations in the angle between the two wings enable different ligands to induce different signaling complexes of RET. This mechanism is reminiscent of the 'biased agonism' paradigm that is well-established for G-protein-coupled receptors (GPCRs), where multiple ligands of the same receptor can preferentially activate different downstream signaling pathways by inducing distinct activation kinetics or conformations of the receptor (*Wacker et al., 2017*). Recent studies have shown that some single-pass transmembrane receptors such as interferon-$^\gamma$ receptor and the erythropoietin receptor are also able to mediate biased signaling for certain ligands (*Kim et al., 2017*; *Mendoza et al., 2019*). It is possible that the differences in endocytosis between the 2:2:2 versus 4:4:4 complexes of RET enable biased signaling and ultimately lead to profoundly different biological effects in vivo. In line with this idea, a recent study has shown that different ligands of EGFR, while inducing structurally similar active dimers of EGFR, can drive either cell proliferation or differentiation depending on the kinetic stability of the EGFR dimer (*Freed et al., 2017*). In addition to different signaling kinetics, the 4:4:4 complex of RET may signal through unique downstream effectors that the 2:2:2 complex cannot bind. The distinct signaling pathways triggered by different ligands through RET may allow it to control the diverse biological processes such as development and appetite under different contexts.

# Materials and methods

## Key resources table

| Reagent type (species) or resource | Designation | Source or reference | Identifiers | Additional information |
|---|---|---|---|---|
| Antibody | Mouse monoclonal anti-Myc tag | Cell Signaling Technology | Cat# 2276S; RRID: AB_331783 | Dilution 1:1000 |
| Antibody | Rabbit monoclonal anti-p44/42 MAPK (Erk1/2) | Cell Signaling Technology | Cat# 4695; RRID: AB_390779 | Dilution 1:1000 |
| Antibody | Rabbit monoclonal anti-Phospho-p44/42 MAPK (Erk1/2) (Thr202/Tyr204) | Cell Signaling Technology | Cat# 4370; RRID: AB_2315112 | Dilution 1:1000 |
| Antibody | Rabbit monoclonal anti-EEA1(C45B10) | Cell Signaling Technology | Cat# 3288; RRID: AB_2096811 | Dilution 1:1000 |
| Antibody | Goat anti-Mouse IgG H and L (HRP) | Invitrogen | Cat# 31430 | Dilution 1:1000 |
| Antibody | Goat anti-Rabbit IgG H and L (HRP) | Abcam | Cat# ab6721; RRID: AB_955447 | Dilution 1:1000 |
| Antibody | Goat anti-Rabbit IgG H and L (Alexa Fluor 488) | Invitrogen | Cat# A-11034 | Dilution 1:1000 |
| Cell line (*Homo sapiens*) | FreeStyle 293 F | Invitrogen | Cat#R79007 | |
| Cell line (*Homo sapiens*) | HEK293S GnTI- | ATCC | Cat#CRL-3022; RRID: CVCL_A785 | |
| Cell line (*Homo sapiens*) | HEK293 | ATCC | Cat# PTA-4488, RRID: CVCL_0045 | |

*Continued on next page*

*Continued*

| Reagent type (species) or resource | Designation | Source or reference | Identifiers | Additional information |
|---|---|---|---|---|
| Cell line (*Homo sapiens*) | HEK293T | ATCC | Cat#CRL-3216; RRID: CVCL_0063 | |
| Cell line (*Cercopithecus aethiops*) | COS-7 | ATCC | Cat# CRL-1651, RRID: CVCL_0224 | |
| Strain, strain background (*Escherichia coli*) | *E. coli* BL21 (DE3) | New England Biolabs | Cat# C2527 | |
| Strain, strain background (*Escherichia coli*) | XL10-Gold Ultracompetent Cells | Agilent | Cat# 200315 | |
| Strain, strain background (*Escherichia coli*) | MAX Efficiency DH10Bac Competent Cells | Invitrogen | Cat# 10361012 | |
| Chemical compound, drug | isopropyl b-D-thiogalatop yranoside (IPTG) | Fisher Scientific | Cat# BP1620-10 | |
| Chemical compound, drug | Imidazole | Sigma-Aldrich | Cat# I5513 | |
| Chemical compound, drug | L-Arginine | Sigma-Aldrich | Cat# A5006 | |
| Chemical compound, drug | L(-)-Glutathione, oxidized | ACROS Organics | Cat# AC320220050 | |
| Chemical compound, drug | Guanidine-HCl | Thermo Scientific | Cat# 24110 | |
| Chemical compound, drug | L-Glutathione reduced | Sigma-Aldrich | Cat# G4251 | |
| Chemical compound, drug | Urea | RPI | Cat# U20200 | |
| Chemical compound, drug | Puromycin | InvivoGen | Cat# ant-pr-1 | |
| Chemical Compound, drug | Blasticidin | InvivoGen | Cat# ant-bl-1 | |
| Chemical compound, drug | Antibiotic Antimycotic Solution (100×) | Sigma-Aldrich | Cat# A5955 | |
| Chemical compound, drug | Cellfectin II Reagent | Gibco | Cat# 10362100 | |
| Chemical compound, drug | Sodium Butyrate | Sigma-Aldrich | Cat# 303410 | |
| Chemical compound, drug | Halt Protease and Phosphatase Inhibitor Cocktail (100X) | Thermo Scientific | Cat# 78442 | |
| Chemical compound, drug | SuperSignal West Dura Extended Duration Substrate | Thermo Scientific | Cat# 34075 | |

*Continued on next page*

*Continued*

| Reagent type (species) or resource | Designation | Source or reference | Identifiers | Additional information |
|---|---|---|---|---|
| Chemical compound, drug | DAPI (4',6-Diamidino-2-Phenylindole, Dihydrochloride) | Invitrogen | Cat# D1306 | |
| Chemical compound, drug | Alexa Fluor 555 NHS Ester | Invitrogen | Cat# A20009 | |
| Transfected construct (*Homo sapiens*) | pEZT-BM vector | Ryan Hibbs Lab | Addgene plasmid # 74099 | |
| Transfected construct (*Homo sapiens*) | pEZT-RET-ECD-His | This paper | | Materials and methods subsection: Protein expression and purification |
| Transfected construct (*Homo sapiens*) | pEZT-GFRAL-ECD-His | This paper | | Materials and methods subsection: Protein expression and purification |
| Transfected construct (*Homo sapiens*) | pEZT-GFRA1-ECD-His | This paper | | Materials and methods subsection: Protein expression and purification |
| Transfected construct (*Homo sapiens*) | pEZT-GFRA2-ECD-His | This paper | | Materials and methods subsection: Protein expression and purification |
| Transfected construct (*Homo sapiens*) | pEZT-GFRA3-ECD-His | This paper | | Materials and methods subsection: Protein expression and purification |
| Transfected construct (*Escherichia coli*) | pET-28a vector | Novagen | Millipore Cat# 69864 | |
| Transfected construct (*Escherichia coli*) | pET-15b vector | Novagen | Millipore Cat# 69661 | |
| Transfected construct (*Escherichia coli*) | pET-28a-GDF15-WT | This paper | | Materials and methods subsection: Protein expression and purification |
| Transfected construct (*Escherichia coli*) | pET-28a-GDF15-W228E | This paper | | Materials and methods subsection: Protein expression and purification |
| Transfected construct (*Escherichia coli*) | pET-28a-GDF15-Y297E | This paper | | Materials and methods subsection: Protein expression and purification |
| Transfected construct (*Escherichia coli*) | pET-28a-sumo-NRTN-WT | This paper | | Materials and methods subsection: Protein expression and purification |
| Transfected construct (*Escherichia coli*) | pET-28a-sumo-NRTN-R101E/R155E | This paper | | Materials and methods subsection: Protein expression and purification |
| Transfected construct (*Escherichia coli*) | pET-15b-GDNF-WT | This paper | | Materials and methods subsection: Protein expression and purification |

*Continued on next page*

*Continued*

| Reagent type (species) or resource | Designation | Source or reference | Identifiers | Additional information |
|---|---|---|---|---|
| Transfected construct (*Escherichia coli*) | pET-28a-sumo- ARTN-WT | This paper | | Materials and methods subsection: Protein expression and purification |
| Transfected construct (*Homo sapiens*) | pLVX-IRES-Puro | Clontech | Cat#632183 | |
| Transfected construct (*Homo sapiens*) | pLVX-RET-myc-IRES-Puro | This paper | | Materials and methods subsection: Cell-based phosphorylation assay for RET |
| Transfected construct (*Homo sapiens*) | pLVX-myc-GFRAL-WT-IRES-Blast | This paper | | Materials and methods subsection: Cell-based phosphorylation assay for RET |
| Transfected construct (*Homo sapiens*) | pLVX-myc-GFRAL-T261R-IRES-Blast | This paper | | Materials and methods subsection: Cell-based phosphorylation assay for RET |
| Transfected construct (*Homo sapiens*) | pLVX-GFRA2-IRES-Blast | This paper | | Materials and methods subsection: Cell-based phosphorylation assay for RET |
| Software, algorithm | MotionCorr2 | *Zheng et al., 2017* | | http://msg.ucsf.edu/em/software/motioncor2.html |
| Software, algorithm | GCTF | *Zhang, 2016* | | https://www.mrc-lmb.cam.ac.uk/kzhang/Gctf/ |
| Software, algorithm | EMAN2 | *Tang et al., 2007* | | https://blake.bcm.edu/emanwiki/EMAN2 |
| Software, algorithm | RELION | *Scheres, 2012b* | | https://www3.mrc-lmb.cam.ac.uk/relion/index.php/Download_%26_install |
| Software, algorithm | Coot | *Emsley et al., 2010* | | https://www2.mrc-lmb.cam.ac.uk/personal/pemsley/coot/ |
| Software, algorithm | Phenix.refine | *Afonine et al., 2018* | | https://www.phenix-online.org/documentation/reference/refinement.html |
| Software, algorithm | Graphpad prism 7.04 | Graphpad | | https://www.graphpad.com/scientific-software/prism/ |
| Software, algorithm | Fiji | *Schindelin et al., 2012* | | https://imagej.net/Fiji |
| Software, algorithm | μManager | Open Imaging | | https://micro-manager.org/ |
| Software, algorithm | SEDFIT | *Schuck, 2000* | | http://www.analyticalultracentrifugation.com/download.htm |
| Software, algorithm | REDATE | *Zhao et al., 2013* | | http://biophysics.swmed.edu/MBR/software.html |
| Software, algorithm | GUSSI | *Brautigam, 2015* | | http://biophysics.swmed.edu/MBR/software.html |
| Other | Ni Sepharose 6 Fast Flow | GE Healthcare | Cat# 17531802 | |
| Other | Strep-TactinXT Superflow | IBA Lifesciences | Cat# 2-4010-025 | |
| Other | Superdex 200 Increase 10/300 GL | GE Healthcare | Cat# 28990944 | |
| Other | Bolt 4–12% Bis-Tris Plus Gels, 10-well | Invitrogen | Cat# NW04120BOX | |

## Protein expression and purification

All the protein sequence information can be found in *Supplementary file 1*. The human Ret extracellular region (residues 1–635) and GFRα1 (residues 1–426) with a C-terminal His$_8$-tag were sub-cloned into the pEZT-BM vector (*Morales-Perez et al., 2016*). GFRAL (residues 20–352), GFRα2 (residues 24–362) and GFRα3 (residues 32–363) with the alkaline phosphatase (AP) signal peptide fused at the N-terminus and a C-terminal His$_8$-tag were sub-cloned into the same vector. All of these proteins were expressed as secreted proteins in FreeStyle 293 F cells (Invitrogen, #R79007) or HEK293S-GnTI$^-$ cells (ATCC, #CRL-3022) using the BacMam system following the standard protocol (*Morales-Perez et al., 2016*). Briefly, bacmids carrying the target genes was generated by transforming the *E. coli* strain DH10BacY (Geneva Biotech). Baculoviruses were produced by transfecting Sf9 cells with bacmids. Baculoviruses (at 1:50 ratio) were used to infect FreeStyle™ 293 F or HEK293S-GnTI$^-$ cells at the density of $\sim 2 \times 10^6$ cells/ml. Twelve hours later, 3 mM sodium butyrate was supplemented to boost protein expression. Culture medium was collected 3 days after infection by centrifugation at 3500 g at 4°C. Proteins were first captured by Ni$^{2+}$-Sepharose 6 Fast Flow resin (GE healthcare) and then further purified by gel filtration chromatography with a Superdex S200 column (GE healthcare). All of the mutants were expressed and purified in the same manner as the respective wild type proteins. A double strep-tagged version of GFRAL was expressed and purified for the pull-down binding assays as well.

The genes coding the mature part of GDNF (residues 77-end) was inserted into pET-15b. The genes coding the mature part of GDF15 (residues 197-end) were codon-optimized and inserted into pET-28a, respectively. Genes coding the mature part of NRTN (residues 95-end) and ARTN (residues 108-end) were codon-optimized and inserted into a modified pET-28a vector encoding a SUMO-tag following the His$_6$-tag. Proteins expressed from these vectors all contain an N-terminal His$_6$-tag. Proteins were expressed in the *E. coli* strain BL21(DE3) as inclusion bodies and refolded according to the published protocol (*Zhang et al., 2002*), but using reduced glutathione and oxidized glutathione to help forming the disulfide bonds. Briefly, bacteria were cultured in 500 ml Terrific broth (TB) medium at 37°C. Protein expression was induced when the O.D. at 600 nm reached 2 by 1 mM IPTG for 4 hr at 37°C. Inclusion bodies were washed according to the published protocol and were dissolved in 10 ml 7 M Guanidine HCl, 0.15 M reduced glutathione, 30 mM Tris-HCL, 1 mM EDTA, pH 8.0. Solution was further clarified by centrifuge at 40,000 g for 30 min and 0.45 μm syringe filter. Solubilized inclusion bodies were added dropwise to 1 L refolding buffer containing 0.5 M L-arginine, pH 8.0, 0.6 mM oxidized glutathione with stirring for 16 hr at room temperature. The refold solution was then filtered by glass fiber prefilters (Minipore) and exchanged to a buffer containing 25 mM Tris-HCl, 200 mM NaCl, pH 8.0 with a Vivoflow 200 system (Sartorius). The refolded proteins were purified by a combination of Ni-NTA affinity and gel filtration chromatography. Urea at 3 M was added to the purification buffer to improve the solubility of all the GDNF family proteins. All of the purification steps were performed at 4°C. All of the mutants were expressed and purified in the same manners as the respective wild type proteins.

The RET protein was mixed with GDF15 protein and GFRAL protein at the molar ratio of 1:1:1 to form the ternary complex. The proteins mixture was incubated for half an hour before being further purified by gel filtration chromatography. In the gel filtration experiment, the superdex 200 increase 10/300 column was firstly equilibrated with two column volumes of buffer (10 mM Tris buffer with 150 mM NaCl and 1 mM CaCl$_2$ (pH 7.4)) at 4°C. ~300 μl sample was injected into the column. A 96-Well plate was placed at the fraction collector for the sample collection. The peak containing the ternary complex was collected and concentrated to 1 mg/ml for cryo-EM analyses (*Figure 1—figure supplement 1*). The GDNF/GFRα1/RET, NRTN/GFRα2/RET and ARTN/GFRα3/RET complexes were obtained with the same method (*Figure 1—figure supplement 1*).

## Cell lines

293FT, Human embryonic kidney: from Invitrogen, identifier: R70007. HEK293 GNT$^-$, Human embryonic kidney. from ATC, identifier: CRL-3022. HEK293, Human embryonic kidney. from ATC, identifier: PTA-4488. HEK293T, Human embryonic kidney. from ATC, identifier: CRL-3216. COS-7. from ATC, identifier: CRL-1651. *E. coli* BL21(DE3). from New England Biolabs, identifier: C2527. XL10-Gold Ultracompetent Cells. from Agilent, identifier: 200315. MAX Efficiency DH10Bac Competent

Cells. from Invitrogen, identifier: 10361012. The identities of all these cell lines have been authenticated. The mycoplasma contamination testing was performed and shown to be negative.

## EM data acquisition

All cryo-EM grids were prepared by applying 3 µl of protein samples (1 mg/ml) to glow-discharged Quantifoil R1.2/1.3 300-mesh gold holey carbon grids (Quantifoil, Micro Tools GmbH, Germany). Grids were blotted for 5.0 s under 100% humidity at 4℃ before being plunged into liquid ethane using a Mark IV Vitrobot (FEI). Micrographs were acquired on a Titan Krios microscope (FEI) operated at 300 kV with a K2 Summit direct electron detector (Gatan), using a slit width of 20 eV on a GIF-Quantum energy filter. EPU software (FEI) was used for automated data collection following standard FEI procedure. A calibrated magnification of 46,730X was used for imaging, yielding a pixel size of 1.07 Å on images. The defocus range was set from −1.5 µm to −3 µm. Each micrograph was dose-fractionated to 30 frames under a dose rate of 4 e-/pixel/s, with a total exposure time of 15 s, resulting in a total dose of about 50 e-/Å$^2$.

## Image processing

For all the four datasets of the different RET ternary complexes, the image processing was carried out with the same workflow, and the detailed data collection and processing statistic is summarized in *Table 1*. Motion correction was performed using the MotionCorr2 program (*Zheng et al., 2017*), and the CTF parameters of the micrographs were estimated using the GCTF program (*Zhang, 2016*). Initially,~5000 particles were picked with EMAN2 from a few micrographs (*Tang et al., 2007*). All other steps of image processing were performed using RELION (*Scheres, 2012b*; *Scheres, 2012a*). Class averages representing projections in different orientations selected from the initial 2D classification were used as templates for automatic particle picking from the full datasets. Extracted particles were binned three times and subjected to 2D classification. Particles from the classes with fine structural feature were selected for 3D classification. Approximately 20,000 particles were selected to generate the initial mode in RELION. Particles from the 3D classes showing good secondary structural features were selected and re-extracted into the original pixel size of 1.07 Å. 3D refinements with C2 symmetry imposed resulted in 3D

**Table 1.** Cryo-EM data collection and model statistics.

| | GDF15/GFRAL /RET | GDNF/GFRα1/RET | NRTN/GFRα2/RET | ARTN/GFRα3/RET |
|---|---|---|---|---|
| Data collection and processing | | | | |
| Magnification | 46,730 | 46,730 | 46,730 | 46,730 |
| Voltage (kV) | 300 | 300 | 300 | 300 |
| Electron exposure (e-/Å$^2$) | 50 | 50 | 50 | 50 |
| Defocus range (µm) | 1.6–3 | 1.6–3 | 1.6–3 | 1.6–3 |
| Pixel size (Å) | 1.07 | 1.07 | 1.07 | 1.07 |
| Final particle number | *520,480* | 37,098 | 247,157 | 114,344 |
| Map resolution (Å) | 3.7 | 4.4 | 3.4 | 3.5 |
| Map Sharpening *B* factor | −190 | −200 | −140 | −140 |
| Model Refinement Rms deviations | | | | |
| Bonds (Å) | 0.006 | 0.005 | 0.005 | 0.007 |
| Angles (°) | 0.978 | 0.994 | 0.797 | 1.040 |
| Validation | | | | |
| Molprobity score | 1.59 | 2.02 | 1.45 | 1.73 |
| Clashscore | 4.43 | 8.53 | 2.74 | 4.35 |
| Rotamer outliers (%) | 0.12 | 0.41 | 0 | 0.12 |
| Ramachandran plot | | | | |
| Favored (%) | 94.6 | 89.5 | 94.2 | 91.1 |
| Allowed (%) | 5.3 | 10.4 | 5.8 | 8.9 |
| Outliers (%) | 0.1 | 0.1 | 0 | 0 |

DOI: https://doi.org/10.7554/eLife.47650.020

reconstructions to relatively low resolution for all the 4 datasets of the different RET ternary complexes. The density for the outer edge of RET appeared blurred, suggesting relative swinging motions of the two wings in the complexes. To improve the resolution, we performed symmetry expansion and focused refinement as described previously (*Bai et al., 2015*; *Zhou et al., 2015*; *Nguyen et al., 2016*). To do so, all particles in the dataset were duplicated and rotated by 180° according to the C2 symmetry by using the 'relion_particle_symmetry_expand' command. The original and rotated particles were combined to form the symmetry expanded dataset. Subsequently, the density for one RET and one co-receptor from one wing was subtracted from the particles in the symmetry expanded dataset. These operations resulted in a new particle set representing one RET, one co-receptor and the ligand dimer from both the left and right wings of the original dataset. The modified particle set was subjected to another round of 3D refinement with a soft mask around one RET, one co-receptor and ligand dimer, leading to a markedly improved resolution for the entire single wing complex. For the NRTN/GFRα2/RET data, we performed a separate 3D refinement with 2-fold symmetry imposed for the particles corresponding to the 4:4:4 complex, resulting a map at 4.3 Å resolution.

All resolutions were estimated by applying a soft mask around the protein density using the gold-standard Fourier shell correlation (FSC) = 0.143 criterion (*Scheres and Chen, 2012*). It worth mentioning that the resolution of the GDNF/GFRα1/RET complex estimated by the FSC between two half maps (4.4 Å) is not consistent with the one measured by the FSC between the map and model (7.5 Å), suggesting that the resolution of the cryo-EM map is overestimated, presumably due to the preferred orientation issue in this dataset. Nevertheless, the map with clearly resolved secondary structural features provides adequate quality for the rigid body docking and restrained refinement of the model. The symmetry expansion and focused refinement approaches failed to significantly improve the map quality of the GDNF/GFRα1/RET complex. Therefore, we only used the map refined with C2 symmetry imposed for the model building and refinement.

The datasets of the complexes contained a subset of particles of RET without the ligand or coreceptor bound. We obtained a 3D reconstruction of apo-RET by refinement of these particles in the GDF15/GFRAL/RET dataset by a procedure similar to that stated above. The resolution of this map is low, due to the small size of the protein. However, it clearly shows the C-clamp shape of apo-RET that is very similar to RET in the ternary complexes.

We also collected a dataset for the oligomeric peak of apo-RET, but could not obtain a reliable 3D reconstruction due to structural flexibility. Instead, we performed a 2D analysis for this data set showing features of the inhibitory dimer of apo-RET.

## Model building and refinement

Model building of the GDF15/GFRAL/RET complex was initiated by docking the crystal structures of the GDF15/GFRAL complex (PDB ID: 5VZ4) (*Hsu et al., 2017*) and the CLD1/CDL2 domains of RET (PDB ID: 2X2U) (*Kjaer et al., 2010*) into the high-resolution one-wing map from the focused refinement in the program Coot (*Emsley et al., 2010*). CLD3, CLD4 and CRD of RET are built de novo in Coot. While the D1 domain of GFRAL was present in the protein sample, it was completely disordered and invisible in the density map. Density at the bottom of the GDF15 dimer for the linker (Residues 623–635) between the CRD and transmembrane region of RET is weak but clearly visible when the map is contoured at lower levels (*Figure 1—figure supplement 5A*). This part was not included in the model because the map does not show enough features for reliable model building. The model containing the GDF15 dimer, one GFRAL and one RET was manually adjusted in Coot and refined against the map by using the real space refinement module with secondary structure and non-crystallographic symmetry restraints in the Phenix package (*Adams et al., 2010*; *Afonine et al., 2018*). Model geometries were assessed by using Molprobity as a part of the Phenix validation tools and summarized in *Table 1*. The refined model was rotated by 180° according to the 2-fold symmetry in the GDF15 dimer. The two models were combined to construct the entire 2:2:2 GDF15/GFRAL/RET complex, which shows good agreement with the 3D reconstruction of the entire complex. The model building of the 2:2:2 complexes of NRTN/GFRα2/RET and ARTN/GFRα3/RET was initiated by docking the RET model refined above and the crystal structure of the NRTN/GFRα2 (PDB ID: 5MR4) (*Sandmark et al., 2018*) and ARTN/GFRα3 complexes (PDB ID: 2GH0) (*Wang et al., 2006*) into the one-wing map. The rest of the process was the same as that for the GDF15/GFRAL/RET complex. A model containing two 2:2:2 NRTN/GFRα2/RET complexes were built and refined

based on the cryo-EM map of 4:4:4 NRTN/GFRα2/RET complex to generate the entire model of 4:4:4 NRTN/GFRα2/RET complex (*Figure 4—figure supplement 1*). The density for the GDNF/GFRα1/RET complex was relatively poor, but sufficient for manual docking of the individual domains of RET, GFRα1 and the GDNF dimer (PDB ID: 3FUB) (*Parkash and Goldman, 2009*). The entire 2:2:2 GDNF/GFRα1/RET complex was refined against the whole map in Phenix using the same methods as above but with tighter geometric restraints.

## Analytical ultracentrifugation

The sedimentation velocity centrifugation experiments were carried out for both of the two peaks of the RET extracellular region from gel filtration chromatography by using a Beckman Coulter Optima XL-I ultracentrifuge. The centrifugation buffer contained 20 mM HEPES pH 7.4, 300 mM NaCl, 4 mM CaCl$_2$. Proteins (400 µL) at three different concentrations (O.D. at 280 nm of 0.8, 0.25, 0.08) were loaded into the 'sample' side of dual-sectored, charcoal-filled Epon centerpieces, while equal volumes of the centrifugation buffer were loaded into the 'reference' sectors. Sapphire windows were employed to sandwich the centerpieces. Filled cells were loaded into an An50Ti rotor and equilibrated at 20°C for about 2.5 hr under vacuum before centrifugation. Data were acquired at 50,000 rpm at 20°C via absorbance at 280 nm running overnight. Data were analyzed by using the *c(s)* methodology in the program SEDFIT (*Schuck, 2000*). A time-dilation factor was applied to the data using the program REDATE (*Zhao et al., 2013*). The program GUSSI was used to generate the all figures featuring *c(s)* distributions (*Brautigam, 2015*).

## Pull-down assay

Streptavidin conjugated beads (IBA Lifesciences) (10 µl) are incubated with 2 µg GDF15 and 6 µg double strep-tagged GFRAL by head-to-head rotation in 650 µl binding buffer (10 mM HEPES, pH 7.4, 150 mM NaCl, 2 mM CaCl$_2$) for 1 hr at 4°C. Beads were washed with 700 µl binding buffer for three times, and then incubated with12 µg Ret in 650 ul binding buffer for 1 hr at 4°C. Beads were further washed with 700 µl binding buffer for three times and subjected to SDS-PAGE analyses.

## Cell-based phosphorylation assay for RET

The coding sequence for full-length human RET with a C-terminal Myc-tag was inserted into the lentivirus vector pLVX-IRES-Puro with puromycin as the selection marker (Clontech). The coding sequence for human GFRAL (wild-type and the T261R mutant) and GFRα2 with a N-terminal Myc-tag were inserted into the lentivirus vector pLVX-IRES-Blast with blasticidin as the selection marker. HEK293 Cells stably expressing RET and GFRAL were selected with 2 µg/ml puromycin and 10 µg/ml blasticidin for 7 days. Cells were cultured in 6-well plates and then serum-starved for 12 hr before treatment with GDF15 wild-type and mutants at 10 nM for 15 min at 37°C. Cells were lysed in the lysis buffer containing 50 mM Tris pH7.5, 5 mM EDTA and 150 mM NaCl, 1% Triton X-100, and protease and phosphatase inhibitor cocktails (Invitrogen). Cleared lysates were resolved on 4–20% Tris-Glycine PAGE gels and subjected to western blot analyses. Images were acquired by using a ChemiDoc imaging system (Bio-Rad). The primary antibodies against Myc (cat. #2276), ERK1/2 (cat. #4695), phosphor-ERK1/2 (cat. #4370) were purchased from Cell Signaling Technology.

## Immuno-fluorescence imaging

RET endocytosis was analyzed with a protocol similar to that used for plexin in a previous study (*Shang et al., 2017*). Briefly, COS-7 cells stably expressing RET/GFRAL or RET/GFRα2 were seeded on coverslips in 6-well plates and cultured for 16 hr. Cells were washed with PBS three times, and then treated with Alexa Fluor-555 labeled GDF15 or NRTN at 15 nM for 5, 10 or 15 min. Cells were immediately put on ice, washed with ice-cold PBS buffer three times and fixed with 4% (w/v) paraformaldehyde (PFA) for 30 min at room temperature. Cells were permeabilized with 0.05% (w/v) saponin for 5 min and non-specific binding sites were blocked by a quenching buffer (0.01% (w/v) saponin, 2% (w/v) BSA, 0.1% (w/v) lysine in PBS, pH7.4) for 20 min. Cells were stained with rabbit anti-EEA1 antibody over night at 4°C and goat anti-rabbit secondary antibody for 1 hr at room temperature. Cells were imaged with a Plan Apo 60 × 1.4 objective (Nikon) on a spinning-disc confocal system built around a Ti-E Perfect Focus microscope (Nikon) with an EM camera (c9100-13; Hamamatsu) controlled by Micro-Manager software (*Edelstein et al., 2010*). Excitation wavelengths used

were 405 nm (for DAPI), 488 nm (for EEA1), 561 nm (for Alexa Fluor-555 labeled ligands) with the corresponding emission filters 460/50, 525/50 and 595/50, respectively. Confocal images were processed using the Fiji distribution of ImageJ (*Schindelin et al., 2012*). A central slice from the stack of confocal images was chosen for calculating the Pearson's correlation coefficient (PCC) between EEA1 and NRTN or GDF15 by using the Coloc two plugin in Fiji. PCC values were calculated for individual cells defined as regions of interest. P-values were calculated using two-tailed Welch's t-test in Prism 7.04.

## Acknowledgements

We thank Defen Lu, Tao Yue and Eunhee Choi for discussions and technical assistance. Single particle cryo-EM data were collected at the University of Texas Southwestern Medical Center (UTSW) Cryo-Electron Microscopy Facility that is funded by a Cancer Prevention and Research Institute of Texas (CPRIT) Core Facility Support Award (RP170644). We thank D Nicastro and D Stoddard for facility access and data acquisition. XB is supported in part by grants from CPRIT (RR160082) and the Welch foundation (I-1944–20180324). XZ is supported in part by grants from the National Institutes of Health (GM088197 and R35GM130289) and the Welch foundation (I-1702). XB and XZ are both Virginia Murchison Linthicum Scholars in Medical Research at UTSW. JLiou is supported by National Institutes of Health Grant GM113079. JLiou is a Sowell Family Scholar in Medical Research at UTSW.

## Additional information

### Funding

| Funder | Grant reference number | Author |
|---|---|---|
| Cancer Prevention and Research Institute of Texas | RR160082 | Xiao-chen Bai |
| Welch Foundation | I-1944-20180324 | Xiao-chen Bai |
| National Institutes of Health | GM088197 | Xuewu Zhang |
| National Institutes of Health | R35GM130289 | Xuewu Zhang |
| Welch Foundation | I-1702 | Xuewu Zhang |

The funders had no role in study design, data collection and interpretation, or the decision to submit the work for publication.

### Author contributions

Jie Li, Guijun Shang, Yu-Ju Chen, Chad A Brautigam, Data curation, Formal analysis, Writing—original draft, Writing—review and editing; Jen Liou, Data curation, Formal analysis, Supervision, Writing—original draft, Writing—review and editing; Xuewu Zhang, Conceptualization, Data curation, Formal analysis, Supervision, Funding acquisition, Validation, Writing—original draft, Writing—review and editing; Xiao-chen Bai, Conceptualization, Data curation, Formal analysis, Supervision, Funding acquisition, Writing—original draft, Writing—review and editing

### Author ORCIDs

Jie Li https://orcid.org/0000-0002-1059-280X
Guijun Shang http://orcid.org/0000-0002-0187-7934
Chad A Brautigam http://orcid.org/0000-0001-6563-1338
Jen Liou http://orcid.org/0000-0003-1546-3115
Xuewu Zhang https://orcid.org/0000-0002-3634-6711
Xiao-chen Bai https://orcid.org/0000-0002-4234-5686

### Decision letter and Author response

Decision letter https://doi.org/10.7554/eLife.47650.050
Author response https://doi.org/10.7554/eLife.47650.051

# Additional files

## Supplementary files

• Supplementary file 1. The sequences of the proteins used in this work.
DOI: https://doi.org/10.7554/eLife.47650.021

• Transparent reporting form
DOI: https://doi.org/10.7554/eLife.47650.022

## Data availability

Cryo-EM maps and the corresponding models of RET/co-receptors/ligands complexes have been deposited in EMDB and PDB under the accession codes EMD-20572/EMD-20573/EMD-20575/EMD-20576/EMD-20577/EMD-20578/EMD-20579/EMD-20580 and 6Q2J/6Q2N/6Q2O/6Q2R/6Q2S, respectively. All data generated or analysed during this study are included in the manuscript and supporting files. Source data files have been provided for Figure 5.

The following datasets were generated:

| Author(s) | Year | Dataset title | Dataset URL | Database and Identifier |
|---|---|---|---|---|
| Jie Li, Guijun Shang, Yu-Ju Chen, Chad A Brautigam, Jen Liou, Xuewu Zhang, Xiao-chen Bai | 2019 | Cryo-EM structure of the RET/GFRa2/NRTN extracellular complex. The 3D refinement was applied with C2 symmetry. | https://www.ebi.ac.uk/pdbe/entry/emdb/EMD-20576 | Electron Microscopy Data Bank, EMD-20576 |
| Jie Li, Guijun Shang, Yu-Ju Chen, Chad A Brautigam, Jen Liou, Xuewu Zhang, Xiao-chen Bai | 2019 | Cryo-EM structure of the RET/GFRa2/NRTN extracellular complex. The 3D refinement was focused on one of two halves with C1 symmetry applied. | https://www.ebi.ac.uk/pdbe/entry/emdb/EMD-20577 | Electron Microscopy Data Bank, EMD-20577 |
| Jie Li, Guijun Shang, Yu-Ju Chen, Chad A Brautigam, Jen Liou, Xuewu Zhang, Xiao-chen Bai | 2019 | Cryo-EM structure of the RET/GFRa2/NRTN extracellular complex in the tetrameric form | https://www.ebi.ac.uk/pdbe/entry/emdb/EMD-20578 | Electron Microscopy Data Bank, EMD-20578 |
| Jie Li, Guijun Shang, Yu-Ju Chen, Chad A Brautigam, Jen Liou, Xuewu Zhang, Xiao-chen Bai | 2019 | Cryo-EM structure of the RET/GFRa3/ARTN extracellular complex. The 3D refinement was applied with C2 symmetry. | https://www.ebi.ac.uk/pdbe/entry/emdb/EMD-20579 | Electron Microscopy Data Bank, EMD-20579 |
| Jie Li, Guijun Shang, Yu-Ju Chen, Chad A Brautigam, Jen Liou, Xuewu Zhang, Xiao-chen Bai | 2019 | Cryo-EM structure of the RET/GFRa3/ARTN extracellular complex. The 3D refinement was focused on one of two halves with C1 symmetry applied. | https://www.ebi.ac.uk/pdbe/entry/emdb/EMD-20580 | Electron Microscopy Data Bank, EMD-20580 |
| Jie Li, Guijun Shang, Yu-Ju Chen, Chad A Brautigam, Jen Liou, Xuewu Zhang, Xiao-chen Bai | 2019 | Cryo-EM structure of the extracellular complex of RET/GFRAL/GDF15 | https://www.rcsb.org/structure/6Q2J | RCSB Protein Data Bank, 6Q2J |
| Jie Li, Guijun Shang, Yu-Ju Chen, Chad A Brautigam, Jen Liou, Xuewu Zhang, Xiao-chen Bai | 2019 | Cryo-EM structure of the RET/GFRa1/GDNF extracellular complex | https://www.rcsb.org/structure/6Q2N | RCSB Protein Data Bank, 6Q2N |
| Jie Li, Guijun Shang, Yu-Ju Chen, Chad A Brautigam, Jen Liou, Xuewu Zhang, Xiao-chen Bai | 2019 | Cryo-EM structure of the RET/GFRa2/NRTN extracellular complex. | https://www.rcsb.org/structure/6Q2O | RCSB Protein Data Bank, 6Q2O |
| Jie Li, Guijun Shang, Yu-Ju Chen, Chad A Brautigam, Jen Liou, Xuewu Zhang, Xiao-chen Bai | 2019 | Cryo-EM structure of the RET/GFRa2/NRTN extracellular complex in the tetrameric form | https://www.rcsb.org/structure/6Q2R | RCSB Protein Data Bank, 6Q2R |

| Jie Li, Guijun Shang, Yu-Ju Chen, Chad A Brautigam, Jen Liou, Xuewu Zhang, Xiao-chen Bai | 2019 | Cryo-EM structure of the RET/GFRa3/ARTN extracellular complex. | https://www.rcsb.org/structure/6Q2S | RCSB Protein Data Bank, 6Q2S |
| Jie Li, Guijun Shang, Yu-Ju Chen, Chad A Brautigam, Jen Liou, Xuewu Zhang, Xiao-chen Bai | 2019 | Cryo-EM structure of the RET/GFRAL/GDF15 extracellular complex. | https://www.ebi.ac.uk/pdbe/entry/emdb/EMD-20572 | Electron Microscopy Data Bank, EMD-20 572 |
| Jie Li, Guijun Shang, Yu-Ju Chen, Chad A Brautigam, Jen Liou, Xuewu Zhang, Xiao-chen Bai | 2019 | Cryo-EM structure of the RET/GFRAL/GDF15 extracellular complex. The 3D refinement was focused on one of two halves with C1 symmetry applied. | https://www.ebi.ac.uk/pdbe/entry/emdb/EMD-20573 | Electron Microscopy Data Bank, EMD-20 573 |
| Jie Li, Guijun Shang, Yu-Ju Chen, Chad A Brautigam, Jen Liou, Xuewu Zhang, Xiao-chen Bai | 2019 | Cryo-EM structure of the RET/GFRa1/GDNF extracellular complex | https://www.ebi.ac.uk/pdbe/entry/emdb/EMD-20575 | Electron Microscopy Data Bank, EMD-20 575 |

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
