## [Decision Letter]

Thank you for submitting your article "Cryo-EM analyses reveal the common mechanism and diversification in the activation of RET by different ligands" for consideration by *eLife*. Your article has been reviewed by three peer reviewers, one of whom is a member of our Board of Reviewing Editors, and the evaluation has been overseen by Cynthia Wolberger as the Senior Editor. The following individuals involved in review of your submission have agreed to reveal their identity: Daniel J Leahy (Reviewer #2); Adrian Whitty (Reviewer #3).

The reviewers have discussed the reviews with one another and the Reviewing Editor has drafted this decision to help you prepare a revised submission.

Summary:

Li et al., report 4 cryo-EM structures of the RET extracellular region (ECD) complexed with different co-receptors and ligands (GDNF/GFRa1. NTN/GFRa2, ART/GFRa3, and the recently reported GDF15/GFRaL). These structures represent the first views of complete RET ECD/co-receptor/ligand complexes and are highly revealing. The lack of a full structure of RET itself has so far hampered detailed structure-function studies on this important signaling receptor, and by elucidating the structures of RET and of these four RET/Ligand/Co-receptor complexes, this manuscript substantially advances our knowledge and understanding of how RET is activated – with implications for the many other growth factor and cytokine receptors that interact with multiple ligands and/or co-receptors. One very interesting feature of this receptor system is the fact that the individual ligand/co-receptor complexes have significantly different geometries that lead to distinct dimeric arrangements/orientations in the structures summarized in Figure 1B of the manuscript. The authors describe the interactions between the components well, and validate the various ligand/receptor interactions reasonably with mutational studies. The structures seem to confirm the bipartite interactions previously inferred from lower resolution EM studies of GDNF/GFRa1 by McDonald's group, arguing that the CLDs of RET primarily interact with the co-receptor, but the CRD (for which the first structure is reported here) directly engages the ligand.

The observation of higher-order receptor clusters is potentially important, because there has been speculation in the literature that formation of higher-order complexes could be an important feature of many receptor signaling mechanisms – but this has proven to be a difficult question to probe experimentally. Here, the authors observe a 4:4:4 (vs. 2:2:2) complex of NRTN/GFRa2/RET in their EM studies, and argue that mutations at the interface between the 2:2:2 subunits in the 4:4:4 complex result in an enhanced level of signaling and a depressed rate of endocytosis (see comments below). Interestingly, owing to the variable "bat wing" angles within the 2:2:2 complexes with different ligands, not all RET/GFRa/ligand complexes are compatible with formation of 4:4:4 complexes. Thus, the authors may have uncovered an interesting way for selective formation of higher-order receptor oligomers to modulate signaling responses. The authors also provide some evidence that the unbound RET extracellular domain may self-associate to form an inactive homodimer – with the regions of RET that mediate this homotypic interaction indicating that binding of ligand/co-receptor would be expected to compete with RET homodimer formation.

One general criticism is that the authors do not really link their work to previous suggestions from the McDonald lab that the CRD is a major driver of dimerization – although this is implied in the text (and in Figure 1—figure supplement 5A/B). Although the structural work is strong, and documents a variety of RT complexes with its ligands and their co-receptors, several functional aspects of the study are weak. Moreover, there is a surprising focus on some observations that are not well supported as being relevant – at the cost of a better contextualization of the work in the field.

Essential revisions:

1) Is there evidence in the literature for differential responses in the same cells to different RET/GFR ligands?

2) In subsection “Higher-order oligomerization regulates RET signaling and endocytosis”, the authors state that mutation of RET to block higher-order complex formation "markedly reduced NRTN-induced ERK phosphorylation", shown in Figure 5A. Based on the figure provided, I would say that the effect is much more subtle than this description implies. Unless the effect is more striking in the publication-quality figure, it is important that the authors include densitometry analysis and appropriate statistics on the three biological replicates to show that the effect is robust and reproducible.

3) Similarly, the micrographs presented in the manuscript do not seem to show internalization of the mutated NTRN, despite what the EEA1 colocalization graph suggests. Where are the punctae for mutated NRTN? Perhaps it is a micrograph quality issue, but this is just not convincing. This further reduces the extent to which the section on higher-order oligomers is unconvincing, and needs significant additional experimental work and quantitation if the current conclusions are to stand.

4) The authors note that the membrane-contacting termini of the four RET receptors in the higher-order RET tetrameric complex do not lie in the same plane. Based on this finding, they propose that these higher-order complexes favor regions of the membrane with high positive curvature. This conclusion is highly speculative, and feels like a throw-away explanation – requiring much deeper analysis and stronger experimental support to be compelling. What degree of curvature would be required? Is it known that cell membrane regions exist that show this level of curvature? Are there any published data from studies on other systems that provide a precedent for this phenomenon? Are there other explanations for the geometry of the tetrameric complex if this one proves incorrect? If the authors prefer not to go more deeply into this question, they should instead just say that it is unclear exactly how the tetrameric matric contacts the membrane, with the notion that membrane curvature could be involved as one speculative possibility.

4) The authors should indicate what methods were used to verify the identity, purity, and activity of the purified proteins.

5) Some additional experimental details are required, in Materials and methods section, to truly make the work reproducible by other workers. This is an important issue because refolding the mature parts of GDNF-family proteins into the active form can be difficult, so it is important that the authors describe in detail how all proteins used in the study were cloned, expressed, refolded, and purified. In particular, for the cloning, expression and purification work, the authors should specify what are the modifications from the published protocol that they allude to in subsection “Protein expression and purification”. They should also specify the volumes as well as the compositions of the buffers used for each step, etc. If any proteins required modifications to the protocol used for the other family members, these modifications should be specified. The sequences of all constructs should also be provided. The guide should be: what would another investigator need to know to follow precisely what the authors did and get it to work? Some of this detail can be relegated to Supplementary Information if required.

6) The AUC data suggest protein aggregation. Did the authors undertake equilibrium studies to ask whether this is a normal reversible interaction? Its relevance is difficult to assess.

[Editors' note: further revisions were requested prior to acceptance, as described below.]

Thank you for resubmitting your work entitled "Cryo-EM analyses reveal the common mechanism and diversification in the activation of RET by different ligands" for further consideration at *eLife*. Your revised article has been favorably evaluated by Cynthia Wolberger (Senior Editor), a Reviewing Editor, and three reviewers.

The reviewers feel the manuscript has been improved, but there are some remaining issues that need to be addressed before acceptance, as outlined below.

1) Two of the reviewers remain unconvinced by the data on the effects of NRTN mutations on Erk activation in Figure 5A – which is an important point concerning the functional relevance of higher-order Ret complexes. We recommend that these data be removed (and the conclusions adjusted accordingly), or else presented with proper description of statistical significance in a way that is standard for this type of experiment. Without demonstration of statistical significance, the data should not be included. If the figure is retained, a dose/response analysis should be included, since the mutations could simply be affecting binding affinity.

2) The authors should include the procedure for the gel filtration experiments shown in new Figure 1—figure supplement 1, in Materials and methods section.

3) Please label identify bands in Figure 1—figure supplement 1. If possible, it it would be desirable to show additional characterization data such as SDS-PAGE and analytical SEC chromatograms of each protein by itself, or MS data to show the intact mass of each protein.

---

## [Author Response]

[…] One general criticism is that the authors do not really link their work to previous suggestions from the McDonald lab that the CRD is a major driver of dimerization – although this is implied in the text (and in Figure 5—figure supplement 5A/B). Although the structural work is strong, and documents a variety of RT complexes with its ligands and their co-receptors, several functional aspects of the study are weak. Moreover, there is a surprising focus on some observations that are not well supported as being relevant – at the cost of a better contextualization of the work in the field.

We have added one sentence in the Introduction to highlight the contributions of the McDonald paper regarding the role of the CRD in RET dimerization. We believe that the other concerns refer to the same points as some of the questions in the “essential revisions”, which are answered below.

Essential revisions:1) Is there evidence in the literature for differential responses in the same cells to different RET/GFR ligands?

We thank the reviewers for raising this important question. We didn’t find any direct evidence showing that RET can respond differently to different ligands/co-receptors in the same type of cells. Actually, it has been known that GFRα1-3 show distinct spatial and temporal expression levels at different stages of embryonic development. GDNF/GFRα1/RET, NRTN/GFRα2/RET, and ARTN/GFRα3/RET signaling complexes have overlapping roles in neurodevelopment. Therefore, it is possible that, in the same type of cell, GDNF/GFRα1, NRTN/GFRα2, and ARTN/GFRα3 can trigger similar downstream cellular responses through RET. It also has been shown that there are alternative interactions between these three ligands and co-receptors. For instance, GFRα1, in addition to its prevalent partner GDNF, can interact with NRTN or ARTN to activate RET as well, further suggesting the redundancy of RET signaling. Consistently, the cryo-EM structures of GDNF/GFRα1/RET, NRTN/GFRα2/RET, and ARTN/GFRα3/RET complexes show similar architecture that the two RETs in each complex adopt very wide angles.

By contrast, the newly discovered GDF15/GFRAL pair can trigger metabolic responses and suppress the appetite through activating RET, distinct to the roles of GDNF/GFRα1/RET, NRTN/GFRα2/RET, and ARTN/GFRα3/RET, which are primarily involved in mitogenic pathway. On the other hand, both GFRAL and GFRαs have high expression levels in the hindbrain region; therefore, we speculated that GDF15/GFRAL and other ligands/co-receptors may trigger differential responses in cells in the hindbrain. The remarkable conformational differences of RET caused by different ligands/co-receptors (sharp angle verse wide angle; dimer verse tetramer) may provide a mechanism in defining RET signaling specificity, which could potentially allow generating distinct signaling outputs from the same receptor. Similar ideas have been proposed for the EGFR and TLR4 related signaling pathways (Freed et al., 2017; Piccinini et al., 2016). A sentence is added (subsection “RET can form higher-order oligomeric complexes”) to briefly touch on this point.

2) In subsection “Higher-order oligomerization regulates RET signaling and endocytosis”, the authors state that mutation of RET to block higher-order complex formation "markedly reduced NRTN-induced ERK phosphorylation", shown in Figure 5A. Based on the figure provided, I would say that the effect is much more subtle than this description implies. Unless the effect is more striking in the publication-quality figure, it is important that the authors include densitometry analysis and appropriate statistics on the three biological replicates to show that the effect is robust and reproducible.

As suggested, the quantification results of the western blots are now included in the figure. To show the reproducibility of the results, we also included three biological repeats of the experiments in Figure 5—figure supplement 1. Due to the high levels of variability among different repeats, which is often the case in this type of western blots, we did not do averaging and p-value analyses. However, in the four repeats, the trend of higher phosphorylation levels induced by wild type NRTN compared to the R101E/R155E mutant is very clear. We also toned down the description in the manuscript by removing the word “markedly”.

3) Similarly, the micrographs presented in the manuscript do not seem to show internalization of the mutated NTRN, despite what the EEA1 colocalization graph suggests. Where are the punctae for mutated NRTN? Perhaps it is a micrograph quality issue, but this is just not convincing. This further reduces the extent to which the section on higher-order oligomers is unconvincing, and needs significant additional experimental work and quantitation if the current conclusions are to stand.

We apologize for the low quality images. We failed to notice that the figures were compressed by the server of *eLife*, which led to severely decreased quality of the images and obscured puncta. Puncta are clearly visible in our high-resolution original figure. We have provided the original micrographs in the revised submission. As shown in the high resolution image, most of the fluorescently labelled NRTN WT retain on the cellular membrane, but the NRTN mutant is internalized, and co-localized with EEA1, which supports our conclusion that the tetrameric RET complex reduces clathrin-mediated endocytosis.

4) The authors note that the membrane-contacting termini of the four RET receptors in the higher-order RET tetrameric complex do not lie in the same plane. Based on this finding, they propose that these higher-order complexes favor regions of the membrane with high positive curvature. This conclusion is highly speculative, and feels like a throw-away explanation – requiring much deeper analysis and stronger experimental support to be compelling. What degree of curvature would be required? Is it known that cell membrane regions exist that show this level of curvature? Are there any published data from studies on other systems that provide a precedent for this phenomenon? Are there other explanations for the geometry of the tetrameric complex if this one proves incorrect? If the authors prefer not to go more deeply into this question, they should instead just say that it is unclear exactly how the tetrameric matric contacts the membrane, with the notion that membrane curvature could be involved as one speculative possibility.

We thank th reviewers for this critical comment. We completely agree with this reviewer that we don’t know for certain whether this tetrameric RET complex can alter membrane curvature or not. We therefore toned down this point as suggested.

There are indeed papers in the literature showing that the geometry and size of proteins can regulate vesicle secretion or endocytosis by changing the curvature of membrane, through mechanisms that are distinct from the membrane-bending effects of the BAR domains (Shurer et al., 2019; DeGroot, et al., 2018). We have now added a few sentences to describe these findings and cited these papers in the manuscript. We agree that, to establish the relationship between the RET higher order oligomerization and membrane curvature, we need to perform more in vivo experiments, particularly those involving single-molecule techniques. These experiments are beyond the scope of this particular paper, however.

4) The authors should indicate what methods were used to verify the identity, purity, and activity of the purified proteins.

We used SEC and SDS-PAGE to verify the protein identity and purity. As shown in new Figure 1—figure supplement 1, the ligand/co-receptor/RET co-elute in the SEC, and appear in the earlier fractions, as compared with RET alone. The SDS-PAGE results show that there are no other contaminant proteins in these complexes. The earlier elution time and the multiple peaks of the NRTN/GFRa2/RET, ARTN/GFRa3/RET and GDNF/GFRa1/RET complexes suggest partial formation of higher order oligomeric complex. As we only expressed the ECD domain of RET, we cannot perform the in vitro phosphorylation assays to measure the activity of the purified complex. Nevertheless, we validated our structural models by using the mutagenesis based cellular assays, as shown in Figure 3. All these results consistently support our model.

5) Some additional experimental details are required, in Materials and methods section, to truly make the work reproducible by other workers. This is an important issue because refolding the mature parts of GDNF-family proteins into the active form can be difficult, so it is important that the authors describe in detail how all proteins used in the study were cloned, expressed, refolded, and purified. In particular, for the cloning, expression and purification work, the authors should specify what are the modifications from the published protocol that they allude to in subsection “Protein expression and purification”. They should also specify the volumes as well as the compositions of the buffers used for each step, etc. If any proteins required modifications to the protocol used for the other family members, these modifications should be specified. The sequences of all constructs should also be provided. The guide should be: what would another investigator need to know to follow precisely what the authors did and get it to work? Some of this detail can be relegated to Supplementary Information if required.

Point accepted. We have elaborated the method section about the protein refolding. The sequence of all the constructs can be found in the supplementary file as well as in the EMDB/PDB database. All the plasmids used in this work will be provided on request.

6) The AUC data suggest protein aggregation. Did the authors undertake equilibrium studies to ask whether this is a normal reversible interaction? Its relevance is difficult to assess.

Our AUC results in Figure 6E in general do not betray evidence of wholesale, non-specific, non-reversible aggregation. The evidence for this is the lack of aggregated species in the range 15-30 S and the fact that the three different concentrations have the expected amounts of material, i.e. there was no loss due to aggregation/precipitation. Rather, the data are consistent with the reversible, concentration-dependent formation of a discrete entity sedimenting at about 13.5 S. Hydrodynamic scaling laws suggest that this could be a tetramer or pentamer, with the material between 7-10 S representing intermediate forms of the ultimate oligomer that have fast off rates compared to the time-scale of the sedimentation experiment (i.e. k_off_ > 0.01 s^-1^).

We have revised our text (subsection “Oligomerization of RET in the apo-state”) to make this point more clearly.

[Editors' note: further revisions were requested prior to acceptance, as described below.]

The reviewers feel the manuscript has been improved, but there are some remaining issues that need to be addressed before acceptance, as outlined below.1) Two of the reviewers remain unconvinced by the data on the effects of NRTN mutations on Erk activation in Figure 5A – which is an important point concerning the functional relevance of higher-order Ret complexes. We recommend that these data be removed (and the conclusions adjusted accordingly), or else presented with proper description of statistical significance in a way that is standard for this type of experiment. Without demonstration of statistical significance, the data should not be included. If the figure is retained, a dose/response analysis should be included, since the mutations could simply be affecting binding affinity.

Point accepted. We have completely removed the Figure 5A and the related text in the revised manuscript.

2) The authors should include the procedure for the gel filtration experiments shown in new Figure 1—figure supplement 1, in Materials and methods section.

Point accepted. We have rewritten the procedure of the gel filtration experiments in detail in the revised Materials and methods section.

3) Please label identify bands in Figure 1—figure supplement 1. If possible, it would be desirable to show additional characterization data such as SDS-PAGE and analytical SEC chromatograms of each protein by itself, or MS data to show the intact mass of each protein.

Point accepted. We have labelled each protein band in Figure 1—figure supplement 1, and run the SDS-PAGE for each component by itself. This result is shown in Figure 1—figure supplement 1A.